# Inulin promotes appetite in mice by regulating the gut microbiota under conditions of rapid entry to the plateau

Xiaoli Li[1]*, Shengcai Wu[1], Xiaonan Chen[2], Jian Chen[3‡], Bao Liu[3‡], Xianshuai Liang[1‡]

**1** Food and Nutrition Laboratory, Army Logistics Academy, Chongqing, China, **2** Funtional Food Laboratory, Air Force Logistics Academy, Xuzhou, China, **3** Institute of Medicine and Equipment for High Altitude Region, College of High Altitude Military Medicine, Army Medical University, Chongqing, China

☯ These authors contributed equally to this work.
‡ BL, XL and XL also contributed equally to this work.
* 569912068@QQ.com

## Abstract

The objective of this study was to examine the mechanism of inulin by regulating the gut microbiota under conditions of rapid entry to the plateau stage. Fifty 7-week-old SPF-grade C57BL/6J male mice were used as experimental subjects and analysed. This study compared the structural and functional characteristics of food intake, body weight changes, serum appetite hormone levels, and intestinal flora of mice in different doses of inulin intervention and control groups in two distinct environments: plains and the rush-into plateau. The results demonstrated that inulin influenced the functional characteristics of the gut microbiota of mice in terms of energy production, conversion, carbohydrate transport, and metabolism. Furthermore, inulin enhanced the secretion of appetite hormones, resulting in appetite promotion under acute plateau conditions by increasing the relative abundance of beneficial bacteria. In addition, inulin significantly improved the body weight of mice under plateau conditions, particularly in the mid- and high-dose groups of inulin-treated mice.

## Introduction

Oxygen deficiency at the plateau exerts a deleterious influence on the human body[1–3]. Individuals experience notable alterations in their physiological responses, clinical manifestations, and biochemical markers[4–9], which elicit a range of acute plateau reaction symptoms, including headaches, dizziness, loss of appetite, and fatigue[10,11]. Some studies have found that the most common early reaction is gastrointestinal symptoms, followed by nausea and vomiting caused by a decreased appetite. This decreased appetite can be as high as 60% in some cases[12,13]. Overcoming the adverse effects of decreased appetite after rapid entry into the plateau has become a significant problem that requires urgent resolution.

Inulin[14] is a water-soluble dietary fibre widely used as a functional factor[15]. It provides food for beneficial bacteria in the gut, promotes their growth and activity, and helps maintain a healthy balance of gut flora, which is important for digestion and gastrointestinal health, as evidenced by numerous literature[16–18]. Some studies have demonstrated that the addition of inulin to goat milk ice cream can effectively enhance the number of probiotics such as

**Data availability statement:** All relevant data are within the manuscript and its Supporting Information files.

**Funding:** The author(s) received no specific funding for this work.

**Competing interests:** This paper has no known competing financial interests or personal relationships that could have appeared to influence the work reported in this paper.

bifidobacteria and lactobacilli[19]. Chen Qiongling et al. found that mice exhibited significant differences in the composition of gut microbiota after the intervention of low-molecular-weight inulin through animal experiments. The same effect has been observed in the human gut microbiota[20]. Costa et al. demonstrated that inulin significantly affects the absorption of minerals and trace elements in the body[21]. Currently, research on inulin and the impact of gut microbiota on appetite is primarily concentrated in the field of diabetes treatment and obesity control, focusing on the effect of appetite suppression and satiety enhancement by regulating the gut microbiota[22]. Only a limited number of studies have investigated the effects of a plateau environment on gut microbiota. Moreover, there are no published reports or studies related to appetite improvement under conditions of rapid entry into a plateau. This study investigated the potential for minimising the adverse effects of a hypoxic environment on the gut microbiota, particularly in the context of rapidly entering a plateau. Furthermore, we analysed the mechanism by which inulin promotes appetite via gut microbiota regulation.

## Materials and methods

### Chemicals

Inulin(90% purity) was purchased from Likang Nutritional Foods Co., Ltd (Gansu, China). Mice Neuropeptide Y(NPY, an appetite-stimulating hormone) ELISA Kit and mice Peptide YY(PYY, an appetite-suppressing hormone) ELISA Kit were acquired from Elite Biotechnology Co., Ltd (Wuhan, China). DNA Extraction Kit was obtained from Omega Engineering Co., Ltd. (Connecticut, USA). DNA Polymerase was purchased from Quanshijin Biotechnology Co., Ltd (Beijing, China).

### Animals

Fifty 7-week-old SPF-grade C57BL/6J male mice were acquired from Slake King Da Laboratory Animal Technology Co., Ltd. (Hunan, China). All mice were raised in a 5000m plateau simulation chamber purchased from Feng Lei Aviation Ordnance Co., Ltd. (Guizhou, China) and kept at 20±2 °C, 50±8% relative humidity, on a 12-h light/dark cycle, 310 meters altitude. The animals were housed in five cages (n = 10 per cage) with wood shavings and were fed and watered ad libitum. The study protocol was approved by the Laboratory Animal Welfare and Ethics Committee of the Third Military Medical University. The experimental design, experimental procedures, and methods of animal execution in this study were in accordance with the requirements of animal ethics and welfare (AMUWEC20210997). To minimise pain and distress in the mice, an overdose of anaesthesia consisting of 150 mg/kg of ketamine was administered intraperitoneally, followed by cervical dislocation to ensure complete cessation of life functions.

### Experimental design

The five cages were divided into five groups: plain control (PC), plateau control (HC), high-dose (InH), medium-dose (InM), and low-dose (InL) groups. The concentration of the inulin solution was 0.01 ml/g, and the gavage dosage for each group is presented in Table 1.

The specific dose was calculated by weighing the weight of each mice daily, and all gavage was performed between 9:00 am and 10:00 am. The total duration of the experiment was 15 d. After 14 days of gavage, the HC and intervention groups (InH, InM, and InL) were quickly transferred to the plateau simulation chamber at an altitude of 5,000 m for 24 h, with the same raising conditions as before. The PC group was fed under unchanged environmental

**Table 1. Gavage dosage of each group.**

| Group | Dosages | Time(d) |
|---|---|---|
| PC | 0.01ml/g | 30 |
| HC | 0.01ml/g | 30 |
| InH | 500mg/ml | 30 |
| InM | 250mg/ml | 30 |
| InL | 125mg/ml | 30 |

conditions. Plasma was isolated using a KH80 automatic centrifuge (Xiang Yi Laboratory Instrument Development Co., Ltd.). (Hunan, China).

## Measurement of body weight and food intake

The weights of the mice and their respective feeds were recorded daily throughout the experiment.

## Measurement of appetite hormone content

The NPY and PYY levels in the serum of mice were measured according to the instructions provided with the ELISA kit. Measuring these hormones helps elucidate how inulin modulates appetite via gut microbiota-mediated pathways.

## The diversity of gut microbiota detection

The DNA of the mice faecal samples was extracted according to the instructions provided in the E.Z.N.A. Soil DNA Kit. The purity and concentration of the extracted DNA were measured using an AOELAB V-1200 spectrophotometer made by Xiang yi Instruments Co. Ltd (Shanghai, China), and the integrity of the extracted DNA was assessed using agarose gel electrophoresis with a mass/volume fraction of 1% and an electrophoresis bath voltage of 5 V/cm for approximately 20 min. The PCR-amplified DNA from the faecal samples of mice was purified in accordance with the instructions provided with the AxyPrep DNA Gel Extraction Kit [23]. MiSeq data of esp.genome were constructed according to the instructions of NEXTFLEX Rapid DNA-Seq Kit. Illumina sequencing was performed using the MiSeq PE250 high-throughput sequencing platform. The original FASTQ-format sequences obtained from Illumina sequencing underwent quality control using FAST software[24]. The nucleobase sequences were optimised using the DADA2 algorithm to minimise the interference of PCR amplification and sequencing errors on the sequence results as much as possible. The sequence data obtained through noise reduction optimisation are commonly referred to as ASV[25,26].

## Statistical analysis

The data on mice body weight, food intake, and appetite hormones were analysed using one-way ANOVA with SPSS 24.0, with two decimal places retained. All data are presented as the mean ± SD of the indicated number. Statistical significance was set at $p < 0.05$. Gut microbiota diversity data were analysed using the Meggie BioCloud platform (https://cloud.majorbio.com), the sequencing data are available in S1 Data (Supporting Information).

# Results and discussion

## Results

**Food intake and body weight.**  The daily food intake and body weight statistics of the mice in each group are presented in Table 2.

Following a 14-day period of experimentation in a plain environment, the mean daily food intake of each group is presented in Fig 1.

The mean daily food intake of mice in the InH group was significantly lower than that of the PC group (P < 0.001). Similarly, the mean daily food intake of mice in the InM group was significantly lower than that of the PC group (P < 0.01). In contrast, there was no significant difference between the mean daily food intakes of the mice in the other groups and that of the PC group (Fig 2). In a one-day simulation of rapid entry to the plateau at an altitude of 5,000 m, a significant difference was observed in the average daily food intake between the InH and HC groups (P < 0.05), whereas no significant difference was observed between the InL and HC groups (Fig 3).

Table 2.  Statistics of daily food intake and body weight in mice ( $\bar{x}$ ±s, *n*=10).

| group | daily food intake | | body weight | | |
|---|---|---|---|---|---|
| | 1~14 d | 15 d | 0 d | 14 d | 15 d |
| PC | 2.98±0.29 | 3.02±0.02 | 22.64±0.54 | 24.80±0.93 | 24.99 ± 0.95 |
| HC | 2.85 ± 0.20 | 0.24 ± 0.09 | 22.80 ± 0.61 | 24.72±0.77 | 22.43 ± 0.69*** |
| InH | 2.49 ± 0.21*** | 0.67 ± 0.09# | 22.30 ± 0.45 | 23.31±1.10** | 21.28 ± 0.94***## |
| InM | 2.68 ± 0.10** | 0.59 ± 0.05# | 22.82 ± 0.64 | 24.17±1.19 | 22.00 ± 0.92*** |
| InL | 2.94 ± 0.22 | 0.30 ± 0.06 | 22.91 ± 0.54 | 23.67±0.86* | 21.87 ± 0.99*** |

Note: 1. The asterisk (*) indicates a statistically significant difference (P < 0.05), while the double asterisk (**) indicates a highly statistically significant difference (P < 0.01), and the triple asterisk (***) indicates an extremely statistically significant difference (P < 0.001) between the PC group and the other groups.2. The symbol # indicates a significant difference of P < 0.05, while the symbol ## indicates a highly significant difference of P < 0.01 compared to the HC group.

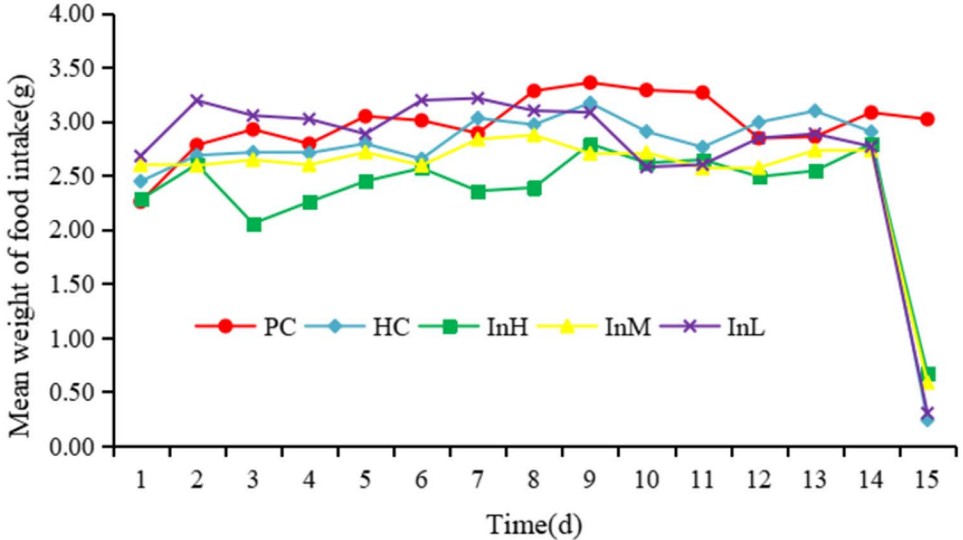

**Fig 1.  Average food intake of each group from 1 to 15 d.**

Figs 1-3 illustrate the variation in food intake of each group of mice on each day during the plains period and after entering the plateau area. The horizontal coordinates of Fig 1 indicate the date and the vertical coordinates the mean weight of food intake, whereas the horizontal coordinates of Figs 2 and 3 indicate the group and the vertical coordinates the mean weight of food intake.

After 14 days in a plain environment, the mice in the PC group gained an average of 2.16 g, while the InH group gained 1.01 g, the InM group gained 1.35 g, the InL group gained

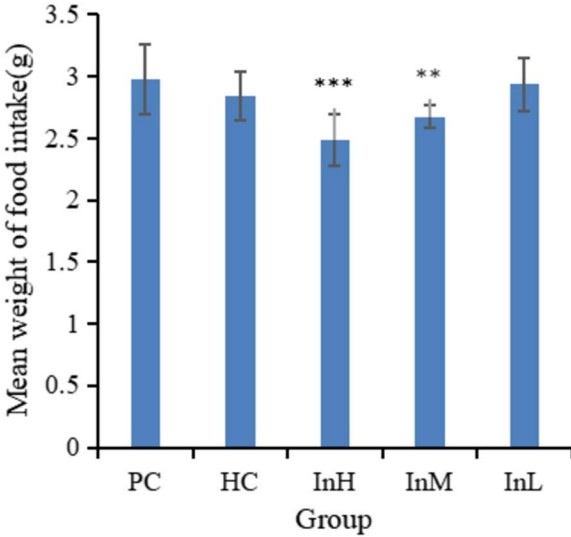

**Fig 2. Average food intake of each group from 1 to 14 d.** Note: The double asterisk (**) indicates a highly statistically significant difference ($P < 0.01$), while the triple asterisk (***) indicates an extremely statistically significant difference ($P < 0.001$) between PC group and the other groups.

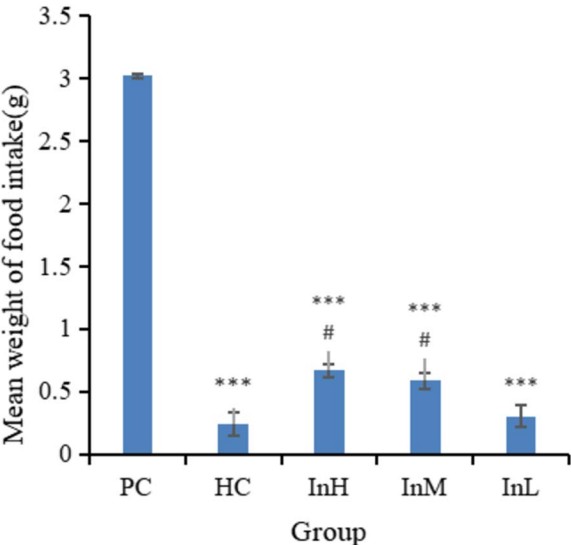

**Fig 3. Average food intake of each group on d 15.** Note: The triple asterisk (***) indicates an extremely statistically significant difference ($P < 0.001$) between the PC group and the other groups. The symbol # indicates a significant difference of $P < 0.05$ compared to the HC group.

0.76 g, and the HC group gained 1.92 g. The InH group weighed more than the PC group after three days. This difference increased as the experiment continued. The InH and PC groups differed significantly in body weight on the 10th day (P < 0.05) and 14th day (P < 0.05). No significant difference in body weight was observed between the HC and PC groups (Figs 4 and 5).

In the one-day plateau experiment, the average body weight of the mice in each group decreased significantly. The InH, InL, and HC groups lost 2.03, 2.17, and 2.29 g, respectively. All groups lost more weight than the PC group (P < 0.001), as shown in Fig 6.

Figs 4-6 illustrate the changes in the daily body weights of the mice in each group in different environments. The horizontal coordinates of Figs 4 and 5 indicate the date and the vertical coordinates the mean weight, while the horizontal coordinates of Fig 6 indicate the group and the vertical coordinates indicate the change in mean weight.

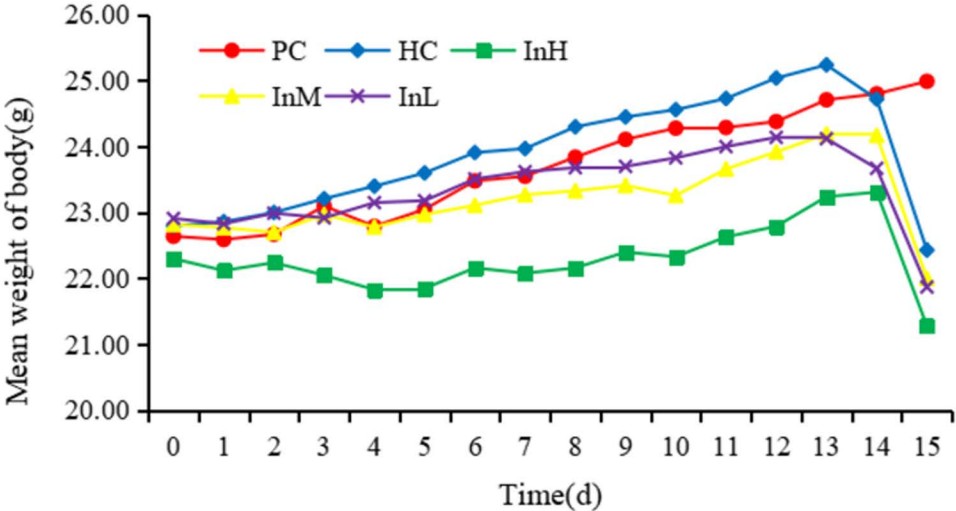

**Fig 4. Average daily body weight of each group.**

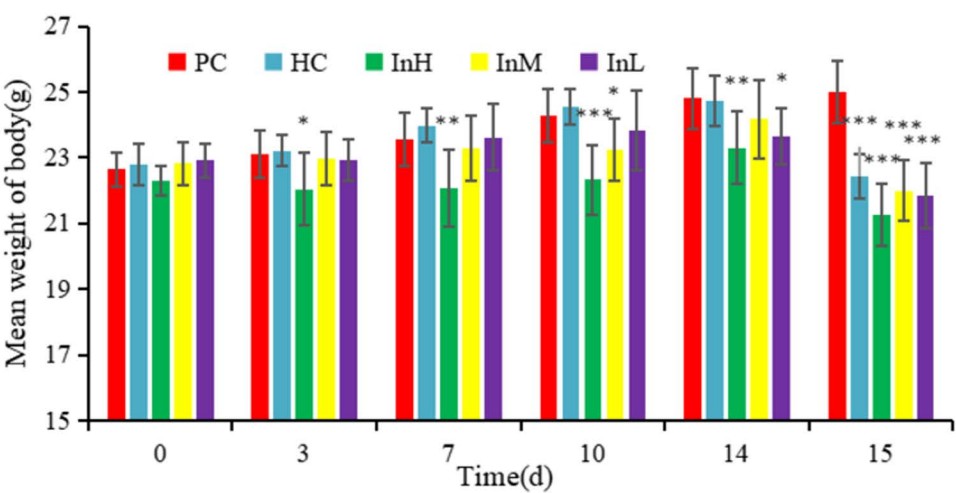

**Fig 5. Average body weight of each group on different days.**

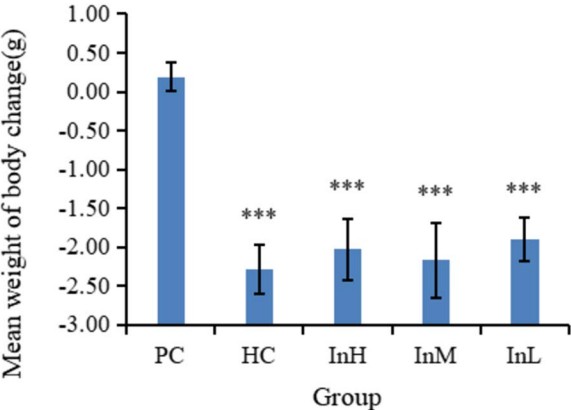

**Fig 6. Changes in average body weight of each group on d 15.** Note: 1. The asterisk (*) indicates a statistically significant difference (P < 0.05), while the double asterisk (**) indicates a highly statistically significant difference (P < 0.01) and the triple asterisk (***) indicates an extremely statistically significant difference (P < 0.001) between PC group and the other groups.

**Serum appetite hormone levels.** NPY and PYY are two important appetite hormones in mice. The average OD value of serum NPY and PYY was calculated using Origin Pro 2021 software. The horizontal coordinate represents the NPY/PYY concentration, and the vertical coordinate represents the OD value. The standard curve was fitted using a four-parameter logistic function [27], as shown in Equations (1) and (2).

$$y = 0.13 + 1.80 / (1 + (x / 154.37)^{0.96}), R^2 > 0.99 \tag{1}$$

$$y = 0.08 + 3.56 / (1 + (x / 1110.32)^{-0.98}), R^2 > 0.99 \tag{2}$$

The OD values of the corresponding samples were substituted into each of the aforementioned regression equations to obtain the serum levels of NPY and PYY, as illustrated in Table 3.

The correlation coefficients of the standard curves for NPY and PYY were all greater than 0.99, indicating that the standard curves were well-fitted and could accurately measure the levels of NPY and PYY in the serum samples of mice. The HC and the other three inulin groups demonstrated a decrease in serum NPY concentration compared with the PC group at the

**Table 3. Changes of appetite hormone in serum of mice under the influence of inulin ( $\bar{x}$ ±s, *n*=10).**

| Group | n | NPY (pg/mL) | PYY (pg/mL) |
|---|---|---|---|
| PC | 10 | 222.78±40.20 | 267.29±17.28 |
| HC | 10 | 151.80±29.75** | 362.78±36.41*** |
| InH | 10 | 197.60±40.11# | 291.21±46.68## |
| InM | 10 | 171.21±41.19* | 314.80±34.09*# |
| InL | 10 | 155.62±37.88* | 331.98±33.50** |

Note: 1. The asterisk (*) indicates a statistically significant difference (P < 0.05), while the double asterisk (**) indicates a highly statistically significant difference (P < 0.01) and the triple asterisk (***) indicates an extremely statistically significant difference (P < 0.001) between the PC group and the other groups.2. The symbol # indicates a significant difference of P < 0.05, while the symbol ## indicates a highly significant difference of P < 0.01 compared to the HC group.

condition of a rapid entry to the plateau. Compared with the HC group, the serum concentrations of NPY were elevated in the three inulin groups after a rapid entry to the plateau under the effect of inulin. A significant difference was observed between the InH and HC groups ($P < 0.05$). In comparison to the PC group, the HC and three inulin groups exhibited elevated serum PYY concentrations after a rapid entry to the plateau. In comparison to the HC group, the serum PYY concentration declined in all three inulin groups following a rapid entry to the plateau under the influence of inulin.

**Sequencing result of gut microbiota.** Polymerase chain reaction (PCR) amplification was conducted on faecal samples from mice, and the resulting products were subjected to quality control analysis. The PCR products were detected by agarose gel electrophoresis, and the target bands exhibited the correct size and concentration, meeting the requisite specifications for subsequent sequencing analysis. Further details are presented in Fig 7.

Fig 7 illustrates the image after PCR amplification of a mice faecal sample and depicts the appropriate dimensions and concentration of the bands. The original, unprocessed gel image corresponding to this analysis is archived in S1 File (Supporting Information).

Following Illumina sequencing and data optimisation, 3,106,453 valid sequences were obtained from the mice faecal samples, with an average sequence length of 420 bp. To ensure the quality of subsequent analysis of the diversity and composition of the bacterial flora, the samples were subjected to sequence levelling in accordance with the minimum number of samples (16,109 sequences per sample). This resulted in the identification of 11,655 ASVs by comparing the removal of chloroplast and mitochondrial sequences. As illustrated in Fig 8, the dilution curve entered a plateau period and rose slowly when the number of sequences exceeded 10,000. The addition of new amplicon sequence variants (ASVs) with an increase in sequencing volume is limited. This indicates that the sequencing volume was sufficient to cover the majority of ASVs in the samples and that the sequencing results can reflect the information of most microbial species in the samples.

Fig 8 indicates that the current sequencing depth is sufficient to cover most microbial species in the sample. The horizontal coordinates indicate the number of read samples, and the vertical coordinates indicate the total number of ASV.

As the sample size increased gradually, a pan-core species analysis was conducted on five groups of samples at the genus level. Changes in the total number of species and the number of core species in each group were obtained. The results are shown in Table 4.

To obtain the gut microbiota of mice at the genus level, the number of pan and core species in the five groups of samples was used to construct pan-core species curves, as shown in Figs 9 and 10. The analysis curve of the Pan-Core species of the mice gut microbiota revealed that the curves gradually flattened as the number of samples increased beyond six. This indicates that with an increase in the number of samples, the number of Pan and Core species of each group tends to flatten, and the number of samples used in this sequencing can be expected to cover the majority of species in the samples.

Figs 9 and 10 depict the variation in the number of pan and core species within each group with respect to sample size. The horizontal coordinates indicate the number of samples, and the vertical coordinates indicate the total number of species.

**Alpha diversity.** An alpha diversity analysis of the gut microbiota in mice was conducted at the genus level, and the results are shown in Table 5.

Among these results, the ace and chao indices, which are positively correlated with species abundance, were lower in the InH and InM groups than in the PC and HC groups. The Shannon index, which was positively correlated with species diversity, exhibited lower values in the InH, InM, and InL groups than in the PC and HC groups, with no significant difference. The Simpson index, which was negatively correlated with species diversity, exhibited higher values

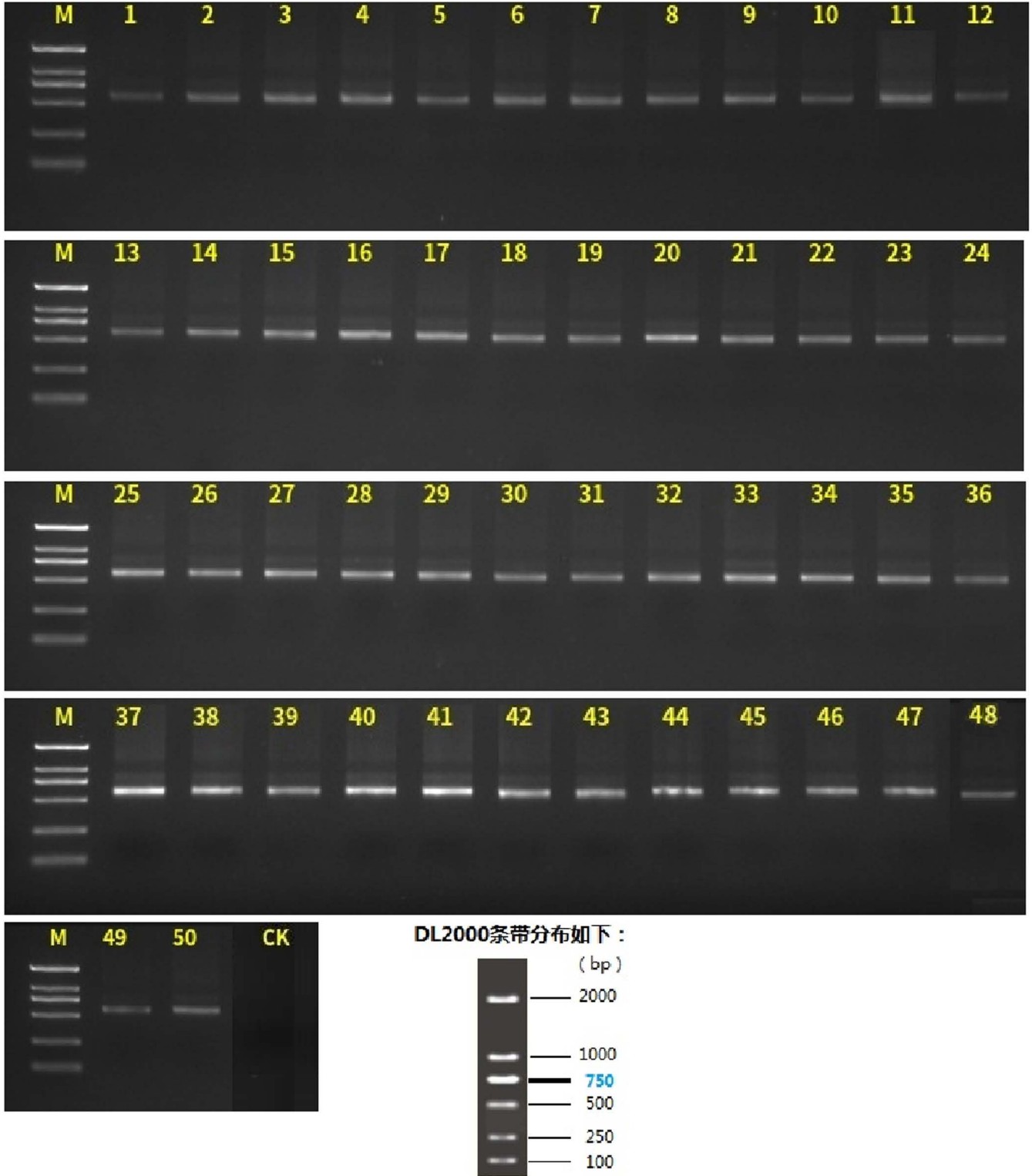

**Fig 7. Agarose gel electrophoresis of PCR products from inulin-treated mice feces.**

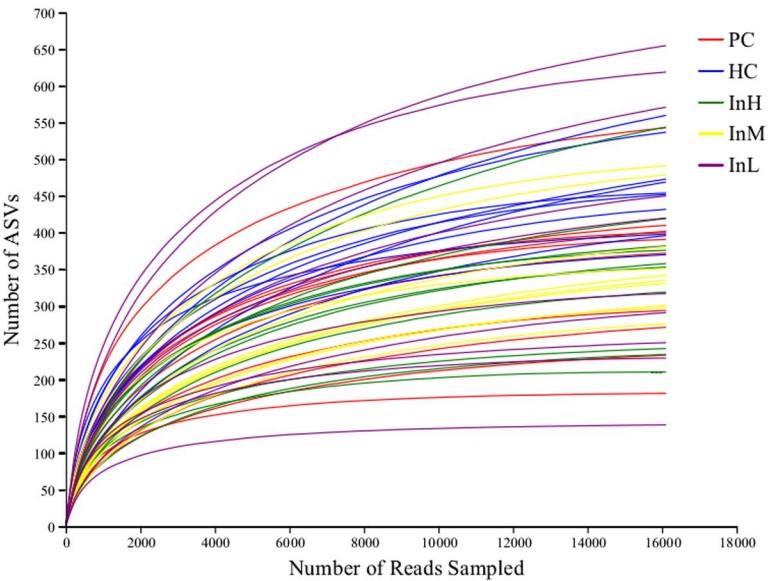

**Fig 8. Rarefaction curves of the gut microbiota from inulin-treated mice feces.**

**Table 4. Pan-core species analysis of gut microbiota at genus level from inulin-treated mice feces. ($n$=10).**

| Group | sample size | Number of species | | | | | | | | | |
|---|---|---|---|---|---|---|---|---|---|---|---|
| | | 1 | 2 | 3 | 4 | 5 | 6 | 7 | 8 | 9 | 10 |
| PC | Pan | 70 | 91 | 105 | 114 | 119 | 125 | 129 | 132 | 134 | 138 |
| | Core | 69 | 49 | 41 | 36 | 33 | 31 | 29 | 28 | 28 | 27 |
| HC | Pan | 80 | 100 | 110 | 117 | 122 | 126 | 130 | 132 | 135 | 137 |
| | Core | 80 | 60 | 51 | 47 | 43 | 40 | 38 | 36 | 34 | 32 |
| InH | Pan*# | 57 | 75 | 86 | 93 | 99 | 104 | 108 | 112 | 114 | 117 |
| | Core# | 59 | 41 | 34 | 31 | 29 | 28 | 27 | 26 | 26 | 25 |
| InM | Pan*## | 59 | 75 | 85 | 92 | 98 | 102 | 105 | 108 | 110 | 112 |
| | Core# | 59 | 43 | 36 | 33 | 31 | 29 | 28 | 28 | 27 | 27 |
| InL | Pan | 76 | 97 | 109 | 115 | 121 | 126 | 130 | 134 | 137 | 139 |
| | Core | 75 | 53 | 42 | 36 | 32 | 29 | 27 | 25 | 24 | 23 |

Note: 1. The asterisk (*) indicates a statistically significant difference ($P < 0.05$). 2. The symbol # indicates a significant difference of $P < 0.05$, while the symbol ## indicates a highly significant difference of $P < 0.01$ compared to the HC group.

in the InH, InM, and InL groups than in the PC group, but exhibited slightly lower values than in the HC group, with no significant difference. The Shannon evenness index, which was positively correlated with species uniformity, was lower in the InH, InM, and InL groups than in the PC group. The species coverage index was lower in the InH, InM, and InL groups than in the PC and HC groups, with no significant difference. The species coverage index of each group was 1, indicating that the sequencing results accurately reflected the actual microbial composition of the samples.

**Beta diversity.** The Bray-Curtis algorithm was employed to calculate the intergeneric distances of the samples, generating a beta diversity distance matrix. The UPGMA algorithm was then utilised to display the species composition of each group and analyse the hierarchical

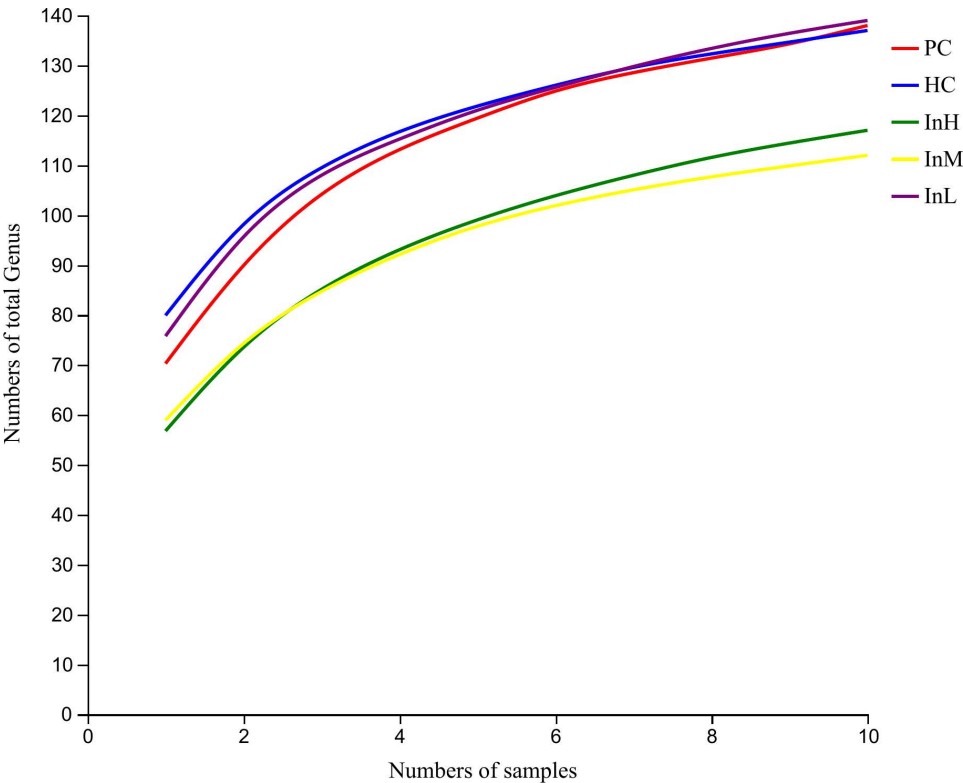

**Fig 9. Genus-level Pan analysis of inulin-treated mice gut microbiota.**

clustering. The hierarchical clustering tree diagram of the samples and the heatmap diagram of the sample distances are presented in Figs 11 and 12.

Figs 11 and 12 depict the variations in the samples of the groups under the influence of inulin. The horizontal coordinates of Fig 11 represent the structural differences in the microbiological composition of the samples, whereas the vertical coordinates indicate the sample numbers. Fig 12 shows the distances between the samples.

The hierarchical clustering tree diagram indicated that the HC group samples exhibited the greatest divergence from the PC group, which was concentrated at the upper and lower ends of the tree diagram. This indicates a significant divergence in microbial diversity between the two sample groups. The samples of the InM group exhibited a closer proximity to the PC group than those of the InH group, indicating that the microbial diversity of the InM group differed to a lesser extent from those of the PC group compared with those of the InH group. The samples of the InL group were located at discrete locations in the tree diagram. The samples of the InL group were distinct in the tree diagram, exhibiting a more differentiated microbial diversity within the group. No evident trend change was observed. The heatmap diagram similarly reflected that the samples of the HC group were more distant from the PC group. Among the three inulin groups, the InM and InL groups were closer to the PC group.

Principal coordinate analysis (PCoA), based on the Bray-Curtis distance matrix, revealed that the microbial diversity at the genus level of each group exhibited the most significant eigenvalues in the coordinate system (Fig 13). The analysis demonstrated that The microbial composition of the InH and InM groups differed significantly from that of the HC group. The microbial composition of the InM group closely resembled that of the PC group.

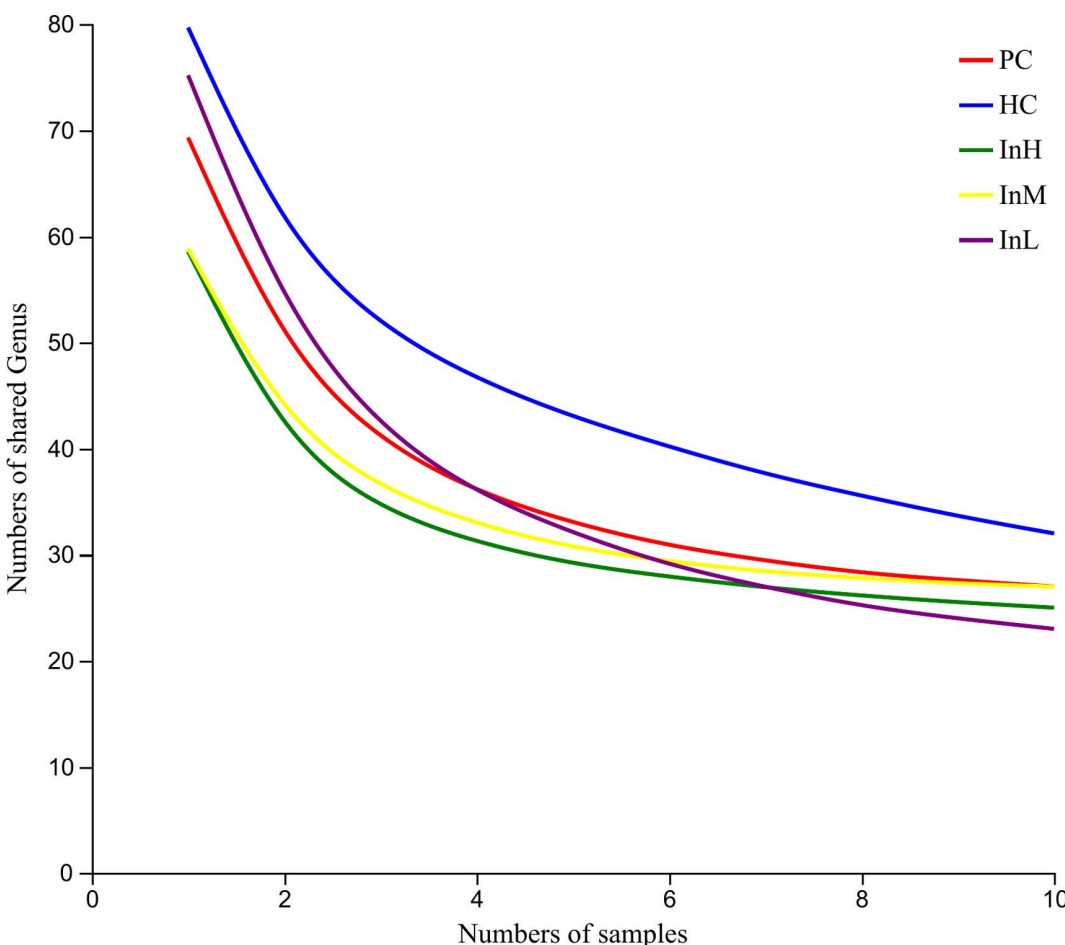

**Fig 10. Genus-level Core analysis of inulin-treated mice gut microbiota.**

**Table 5. Alpha diversity indexes of gut microbiota under the influence of inulin ( $\bar{x}$ ±s, *n*=10).**

| Group | Species abundance index | | Species Diversity Index | | Species uniformity index | Species Coverage Index |
|---|---|---|---|---|---|---|
| | ace | chao | shannon | simpson | shannoneven | coverage |
| PC | 71.49±13.02 | 70.86±13.29 | 2.08±0.25 | 0.23±0.08 | 0.49±0.06 | 1.00±0.00 |
| HC | 82.87±11.58 | 81.95±11.54 | 2.01±0.38 | 0.28±0.11 | 0.46±0.08 | 1.00±0.00 |
| InH | 60.67±10.47[#] | 63.3±14.39[#] | 1.83±0.17 | 0.25±0.04 | 0.45±0.04 | 1.00±0.00 |
| InM | 61.39±8.35[#] | 60.09±8.11[#] | 1.74±0.16 | 0.28±0.04 | 0.43±0.03 | 1.00±0.00 |
| InL | 76.46±19.82 | 76.31±19.99 | 1.99±0.33 | 0.25±0.07 | 0.46±0.05 | 1.00±0.00 |

Note: The symbol # indicates a significant difference of $P < 0.05$ compared to the HC group.

Fig 13 illustrates the similarities and differences in microbial composition between the groups. The percentages indicate the contribution of the principal coordinate components to the difference in sample composition. The scales represent relative distances and have no practical significance.

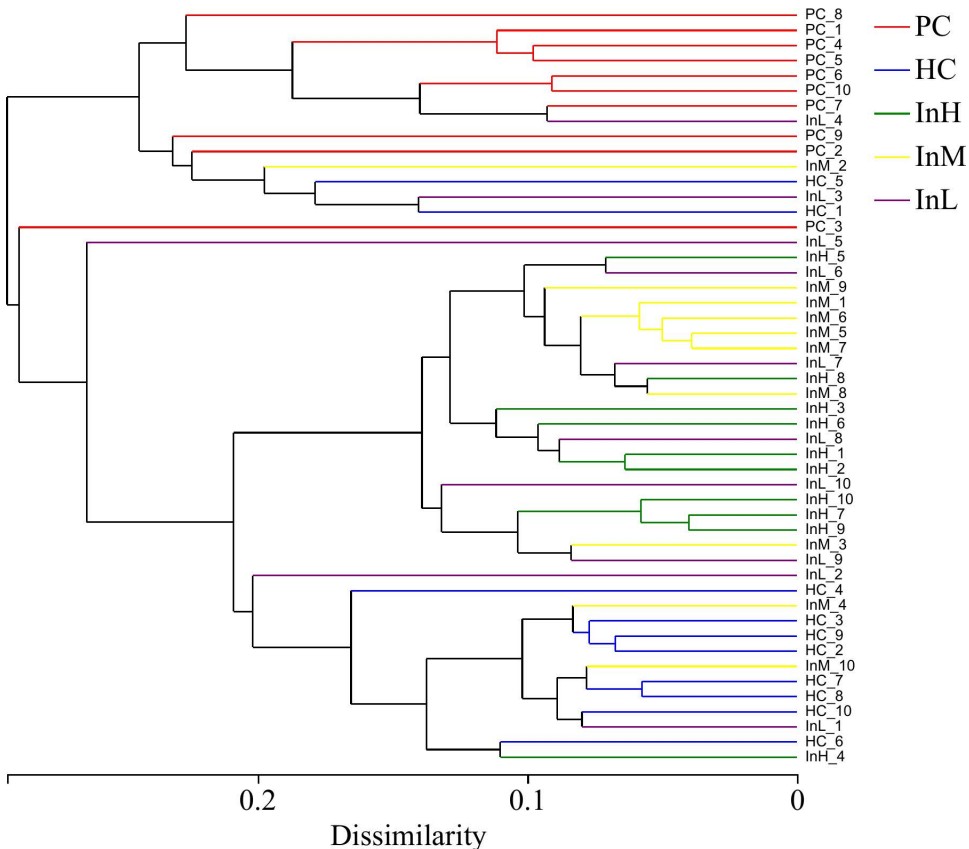

**Fig 11. Clustering tree of gut microbiota from inulin-treated mice feces(n =10).**

**Species composition.** The representative sequences of ASVs were subjected to taxonomic annotation using the Silva138/16s_Bacteria database, which yielded 13 phyla, 21 classes, 48 orders, 83 families, 183 species, and 328 species. Figs 14–18 illustrate the influence of inulin on the relative abundance of flora at the genus level in each group.

The horizontal coordinates of each figure indicate the faecal samples in each group, while the vertical coordinates indicate the composition and relative abundance of the flora. The specific data on the bacteria in each group are shown in Table 6.

**Species differences.** The Kruskal-Wallis rank sum test was employed to analyse the species differences among the five groups of samples at the genus level, based on the relative abundance data of the species. To control the False Discovery Rate and obtain a corrected p-value, the p.adjust function of R statistical software was used to perform false discovery rate (FDR) correction. The relative abundance of several dominant bacterial groups changed following the rapid entry to the plateau under the influence of inulin. The species with increased relative abundance were primarily Bifidobacterium, whereas those with decreased abundance included Akkermansia, Bacillus, Candidatus, and Saccharimonas. Species difference analysis was conducted on the characteristic bacterial groups to evaluate the role of functional foods in regulating the gut microbiota. The relative abundance of bacterial groups was compared (Figs 19–22).

Figs 19-22 illustrate the content and abundance of characteristic flora in each group. The horizontal coordinates of the left figure indicate the percentage of abundance of a species in each group, while the vertical coordinates indicate the grouping category for the two-by-two

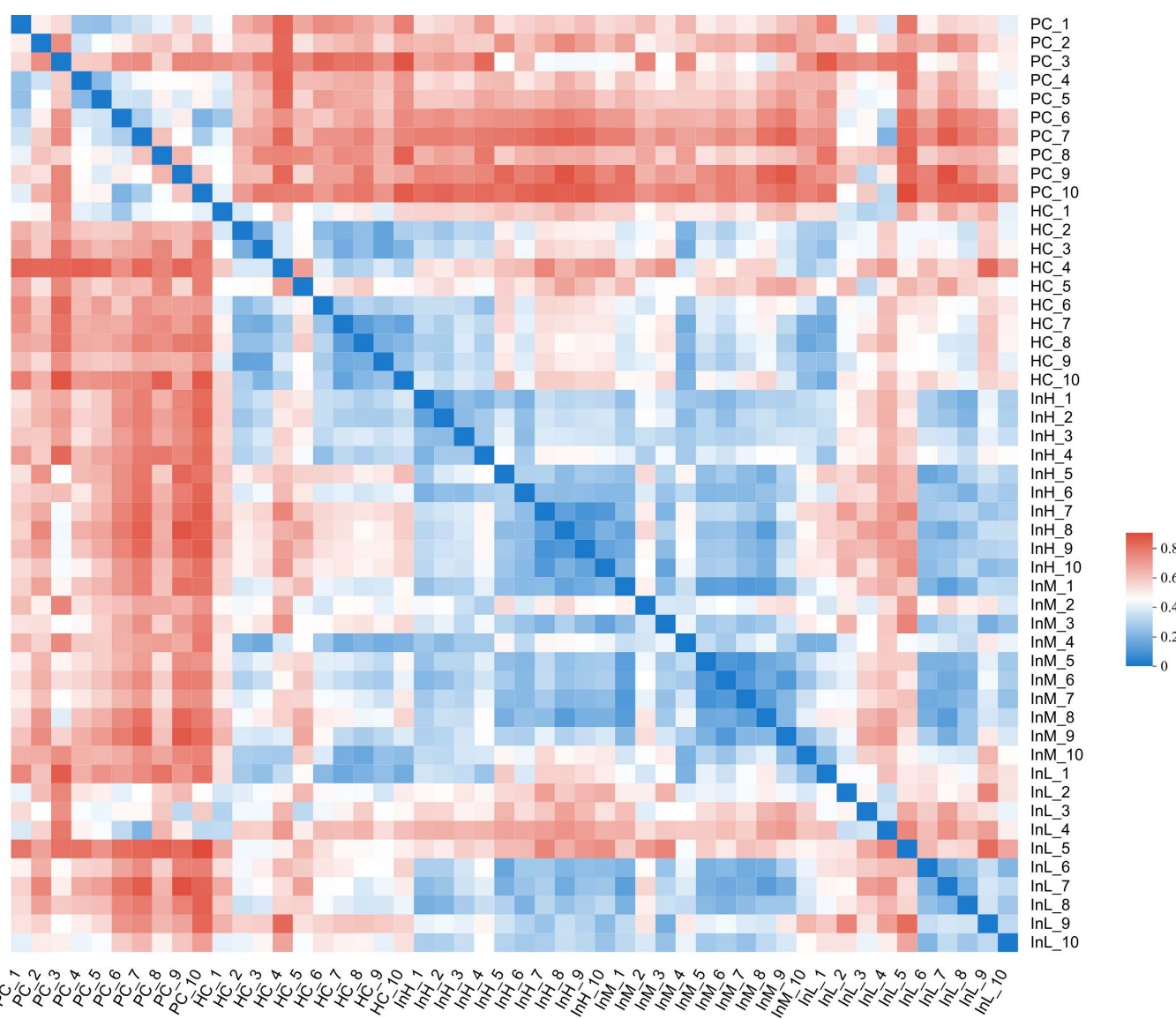

**Fig 12.  Distance heatmap of gut microbiota from inulin-treated mice feces(n =10).**

comparisons. The right figure indicates the proportion of differences in species abundance within the confidence interval, with P-values displayed on the right.

The species difference analysis demonstrated that the relative abundance of Bifidobacterium was lower in the HC group than in the PC group, although this difference was not statistically significant. However, the three inulin groups exhibited higher Bifidobacterium levels than the HC group, with a highly significant difference observed between the InH/InM and HC groups (P < 0.001) and a significant difference between the InL and HC groups (P < 0.05). The relative abundance of Lactobacillus was found to be significantly lower in the HC group compared to the PC group, with an extremely significant difference observed (P < 0.001). In contrast, the InH group exhibited a higher abundance of Lactobacillus, whereas the InM and InL groups exhibited a lower abundance than the HC group. However, no significant

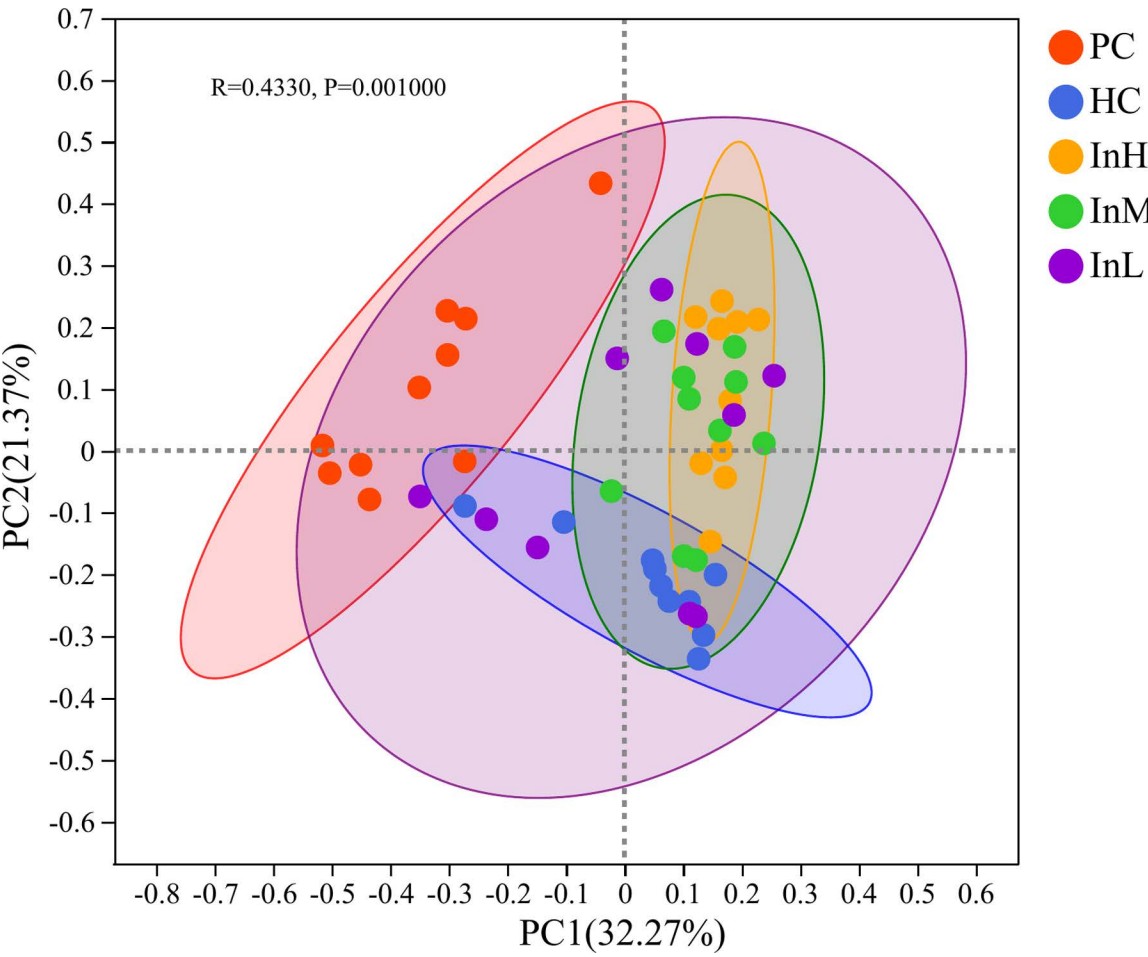

**Fig 13. PCoA analysis of the gut microbiota from inulin-treated mice feces(n =10).**

differences were observed. The relative abundance of Escherichia-Shigella was not detected in the PC or InL groups. In contrast, the other groups exhibited lower abundance below 0.003%, and no significant differences were detected. Regarding the relative abundance of Enterococcus, no detection was observed in the PC and InM groups. The relative abundance of the other groups was less than 0.006% and there were no significant differences between the groups. Clostridium perfringens was not detected in any of the groups.

**Prediction of functional characteristics of mice gut microbiota.** The functional characteristics of the gut microbiota of each group were predicted and analysed using PICRUSt2 software based on the sequencing results of the gut microbiota of the mice. The functional genes obtained from the annotation were categorized according to the Orthologous Groups of proteins(COG) gene function classification, as shown in Fig 23.

Fig 23 provides a visual representation of the species and abundance of gut flora in each group of rats. The horizontal coordinates indicate the groupings, and the vertical coordinates indicate the species of the colonies and their relative abundance.

The COG functions of the mice gut microbiota primarily include translation, transcription, replication, and energy metabolism. In comparison to the PC group, the abundance values of functional features such as energy production and conversion in the HC group, as well as in

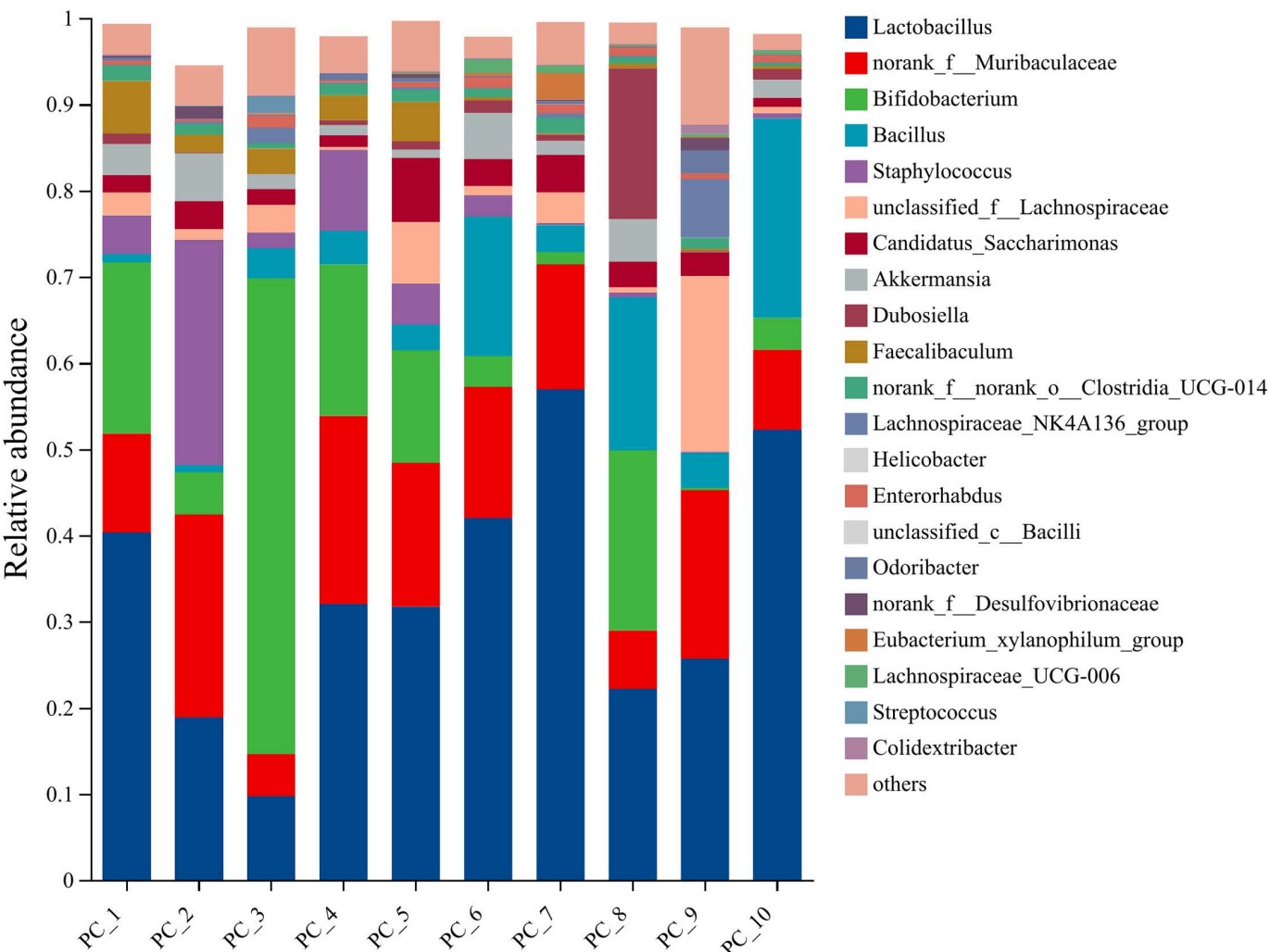

**Fig 14. Genus-level PC group abundance in inulin-treated mice gut microbiota(n =10).**

the inulin groups, were found to be decreased with different degrees. The differences between the groups were statistically significant. Compared to the HC group, the abundance of six functional features, including energy production and conversion, amino acid transport and metabolism, and fat transport and metabolism, was elevated in the InH group. Additionally, the abundance values of transcription, nucleotide transport, and metabolism were elevated. In contrast, the abundance of functional features related to carbohydrate transport and metabolism did not differ significantly. In the InM group, the abundance values of amino acid transport and metabolism and inorganic ion transport and metabolism were elevated and significantly different ($P < 0.01$). The abundance values of nucleotide transport and metabolism and lipid transport and metabolism were elevated and significantly different ($P < 0.05$). The abundance values of the other three functional characteristics were not significantly different from each other. The abundance of functional characteristics in the InL group was not significantly different from that in the HC group.

The abundance values corresponding to the principal functional characteristics of the mice gut microbiota under the influence of inulin are presented in Table 7.

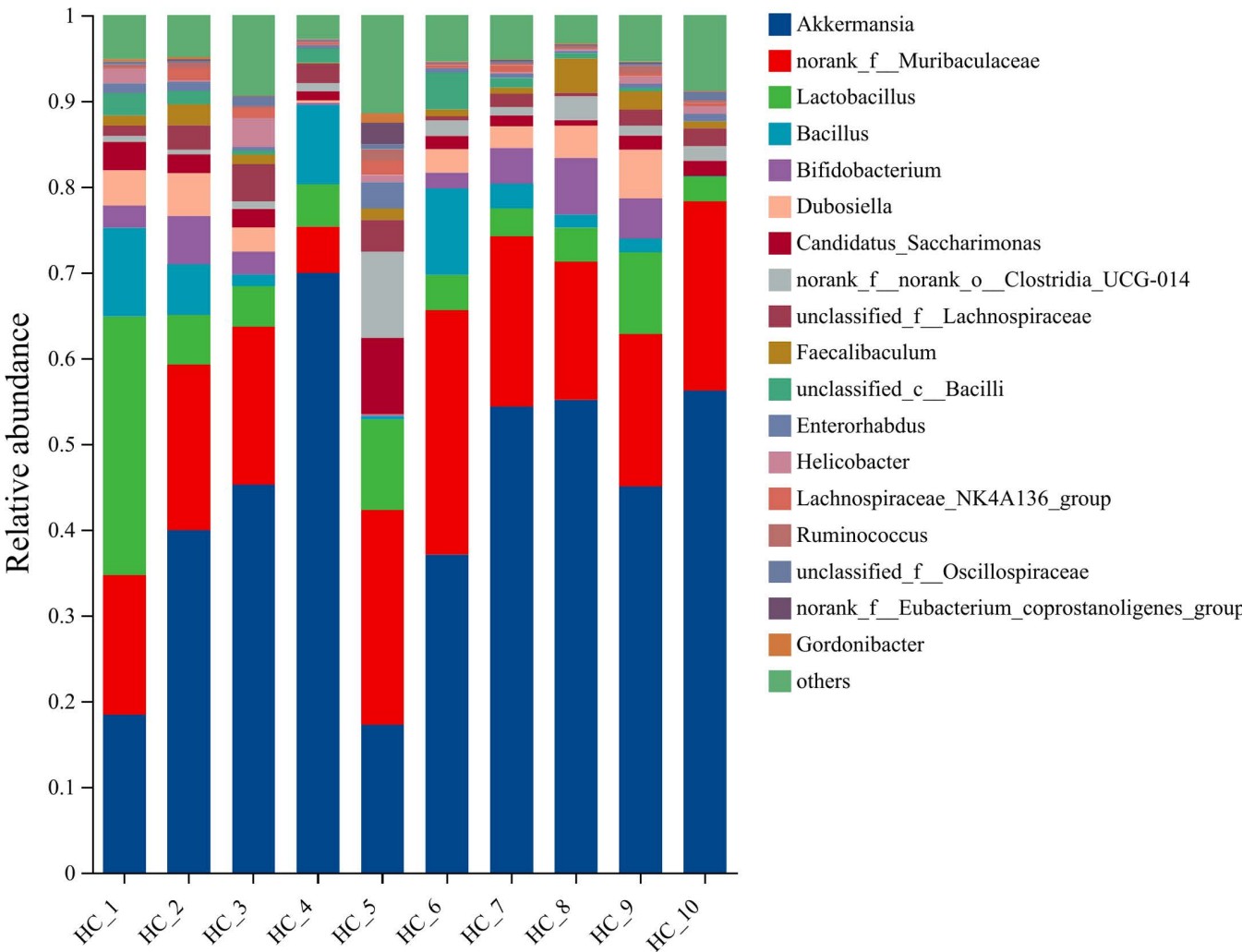

**Fig 15. Genus-level HC group abundance in inulin-treated mice gut microbiota(n =10).**

## Discussion

**The effects of inulin on the food intake of mice.** The average daily food intake of mice in the high-dose inulin group (InH group) and the medium-dose inulin group (InM group) was significantly lower than that of the plain control group (PC group) in the plain environment. First, this result is closely related to the dietary fibre properties of inulin. As a non-digestible carbohydrate, the β-2,1-glycosidic bond in the molecular structure of inulin makes it difficult to hydrolyse and absorb in the intestine. Consequently, inulin can absorb and retain water in the intestinal tract, increasing the volume of the caecum, thus producing a sense of satiety and reducing food intake. Second, fermentation of inulin in the intestinal tract significantly influences food intake in mice. Inulin is fermented by beneficial bacteria (e.g., bifidobacterium and Lactobacillus) in the intestinal tract to produce short-chain fatty acids (SCFA), including acetic, propionic, and butyric acids. These SCFAs not only provide energy to intestinal cells but also regulate appetite and feeding behaviour by influencing the secretion of intestinal hormones. SCFA can activate G protein-coupled receptors within the gut (e.g., GPR41 and GPR43), which are directly or indirectly linked to the central nervous

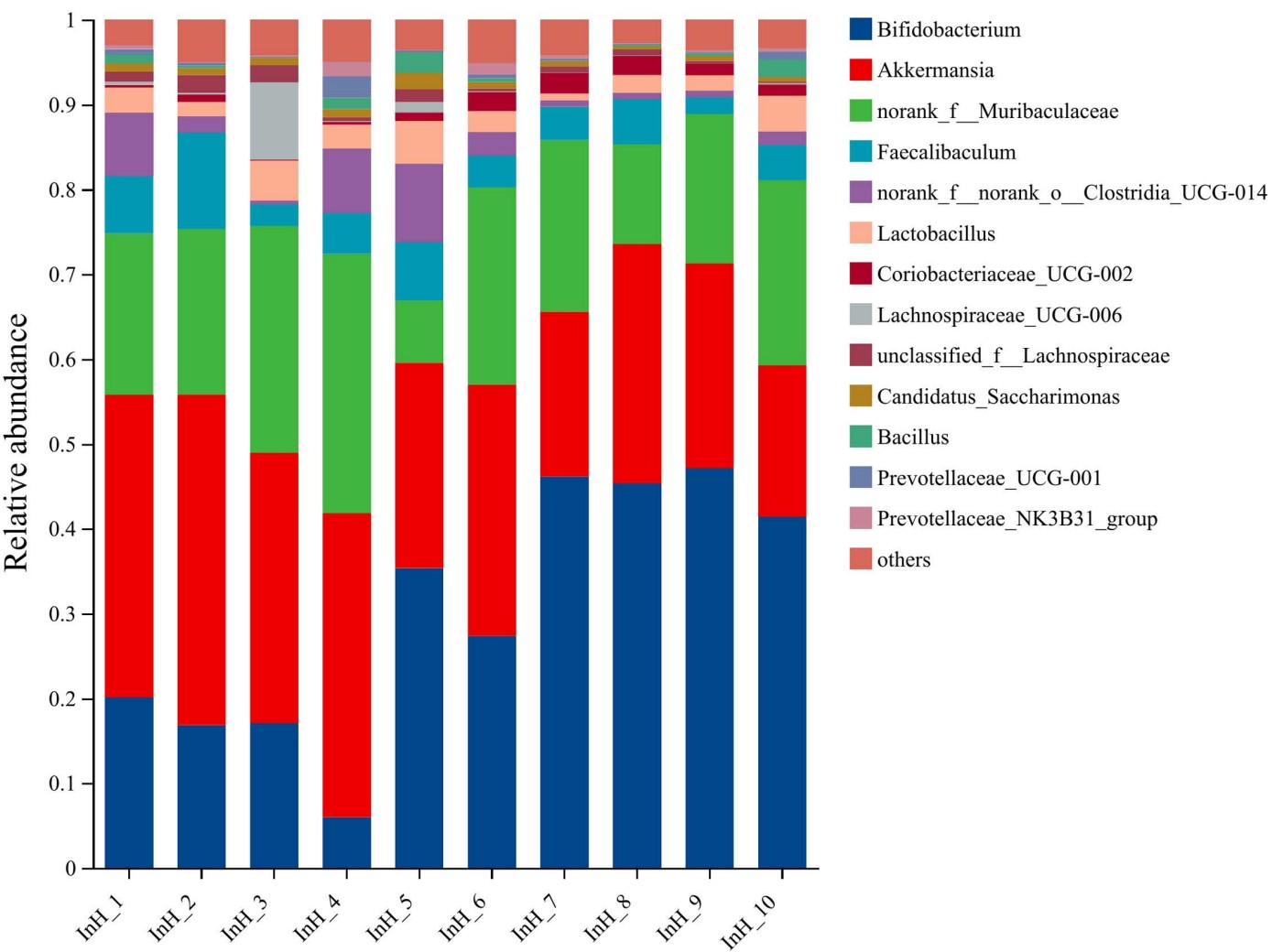

**Fig 16. Genus-level InH group abundance in inulin-treated mice gut microbiota(n =10).**

system (CNS). Upon binding to these receptors, SCFA transmit satiety signals to the brain, thereby suppressing appetite and reducing intake. Furthermore, SCFA can impact the activity of additional appetite regulators, including gastrin-regulating peptide (ghrelin) and insulin-like growth factor (IGF-1), by modulating PH levels in the gut[28]. Third, the dose effect and individual differences can also affect the role of inulin. Notably, there was no significant difference in food intake between the low-dose inulin (InL) and PC groups. This may be attributable to the dose-dependent effects of inulin. It is possible that there is a threshold for the appetite-regulating effect of inulin, whereby only a certain dose range can significantly affect the intake of mice fed a high-fat diet. A low dose of inulin was insufficient to produce a significant amount of SCFA, which in turn was inadequate to exert a notable influence on appetite regulation. Consequently, there was no appreciable alteration in food intake. Furthermore, individual differences are significant factors influencing the impact of inulin. Different mice may exhibit varying degrees of sensitivity to inulin, potentially linked to factors such as intestinal flora structure, metabolic rate, and genetic background. Consequently, future studies should delve deeper into the dose-effect relationship of inulin and the influence of individual differences on its effects.

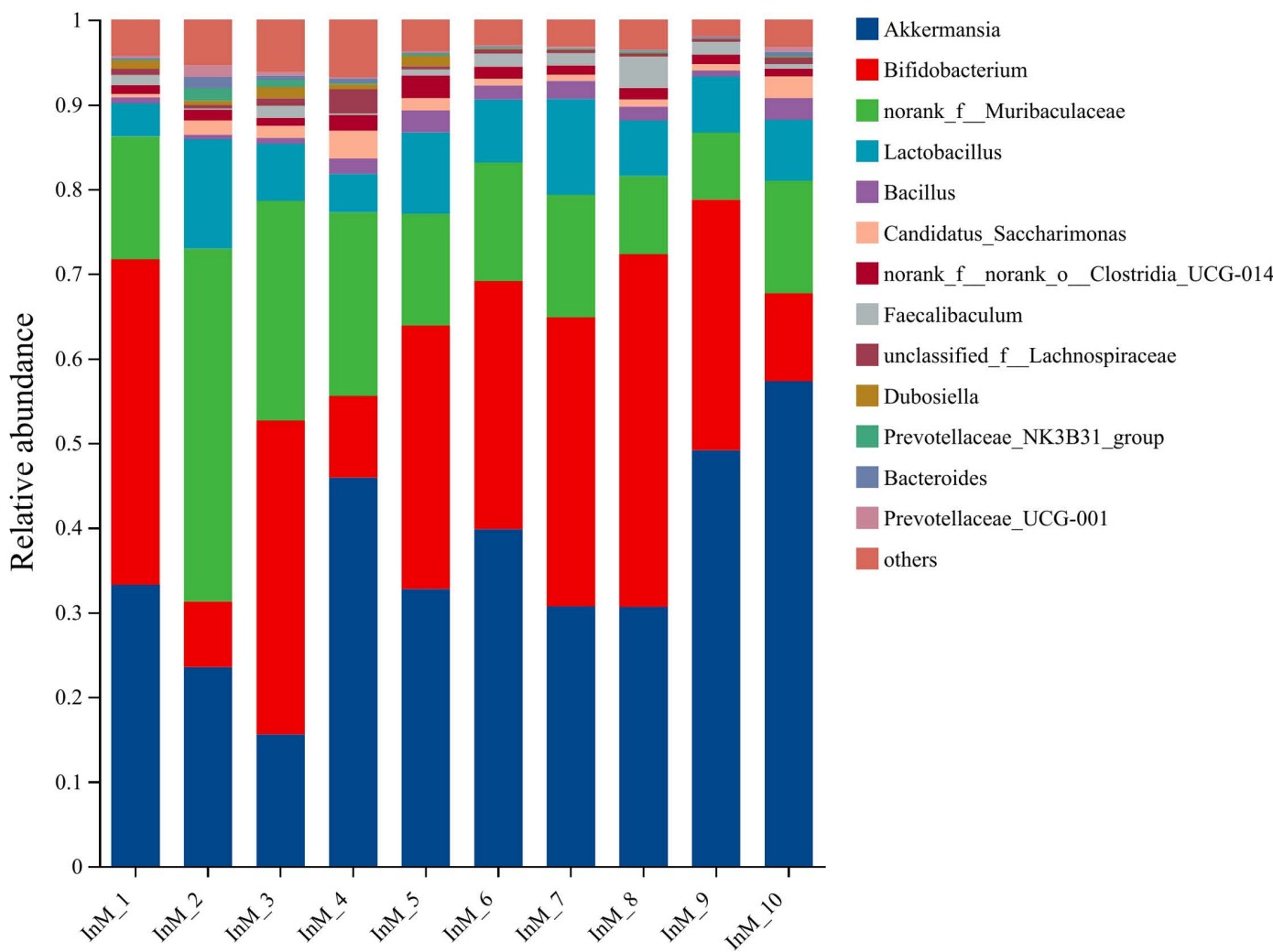

**Fig 17. Genus-level InM group abundance in inulin-treated mice gut microbiota(n =10).**

Overall, inulin stimulated appetite in a plateau environment. When the experimental environment was altered from plain to plateau, a significant reduction in food intake was observed in all mice experimental groups. This was primarily attributable to the combined impact of hypoxia, cold, and elevated energy expenditure associated with the plateau environment[29]. However, mice in the inulin groups (InH, InM, and InL) exhibited a lesser reduction in food intake than those in the plateau control group (HC). This indicates that inulin may have mitigated the inhibitory impact of the plateau environment on the appetite of mice to a certain degree. This alleviating effect may be related to the ability of inulin to improve the structure of intestinal flora and mitigate the negative effects of the plateau environment on intestinal flora. It has been demonstrated that a plateau environment can result in a reduction in the diversity of intestinal flora, a decline in the number of beneficial bacteria, and an increase in the number of harmful bacteria[30]. Inulin maintains equilibrium within the intestinal flora by facilitating the growth and reproduction of beneficial bacteria while simultaneously inhibiting the growth and reproduction of harmful bacteria. This equilibrium helps maintain the typical physiological function of the intestinal tract, thereby mitigating the deleterious effects of the plateau environment on the intestinal tract and appetite.

**Table 6. Specific data of bacteria contained in each group.**

| Group | Total | Unique | Greater | Dominant species |
|---|---|---|---|---|
| PC | 113 | 0 | 21 | Lactobacillus (33.17%), norank_f_Muribaculaceae (14.37%), Bifidobacterium (14.04%), Bacillus (7.64%) |
| HC | 137 | 0 | 18 | Akkermansia (43.86%), norank_f_Muribaculaceae (14.37%), Lactobacillus (8%), Bifidobacterium (2.85%) |
| InH | 117 | 2 | 13 | Bifidobacterium (30.27%), Akkermansia (28.58%), norank_f_Muribaculaceae (19.78%), Faecalibaculum (5.15%) |
| InM | 112 | 4 | 13 | Akkermansia (35.84%), Bifidobacterium (26.92%), norank_f_Muribaculaceae (17.60%), Lactobacillus (7.68%) |
| InL | 139 | 12 | 17 | Akkermansia (28.51%), Bifidobacterium (17.23%), norank_f_Muribaculaceae (15.42%), and Lactobacillus (15.23%) |

Note:Total means the number of total species;Unique means the number of unique species;Greater means the number of species with average relative abundance greater than 1%.

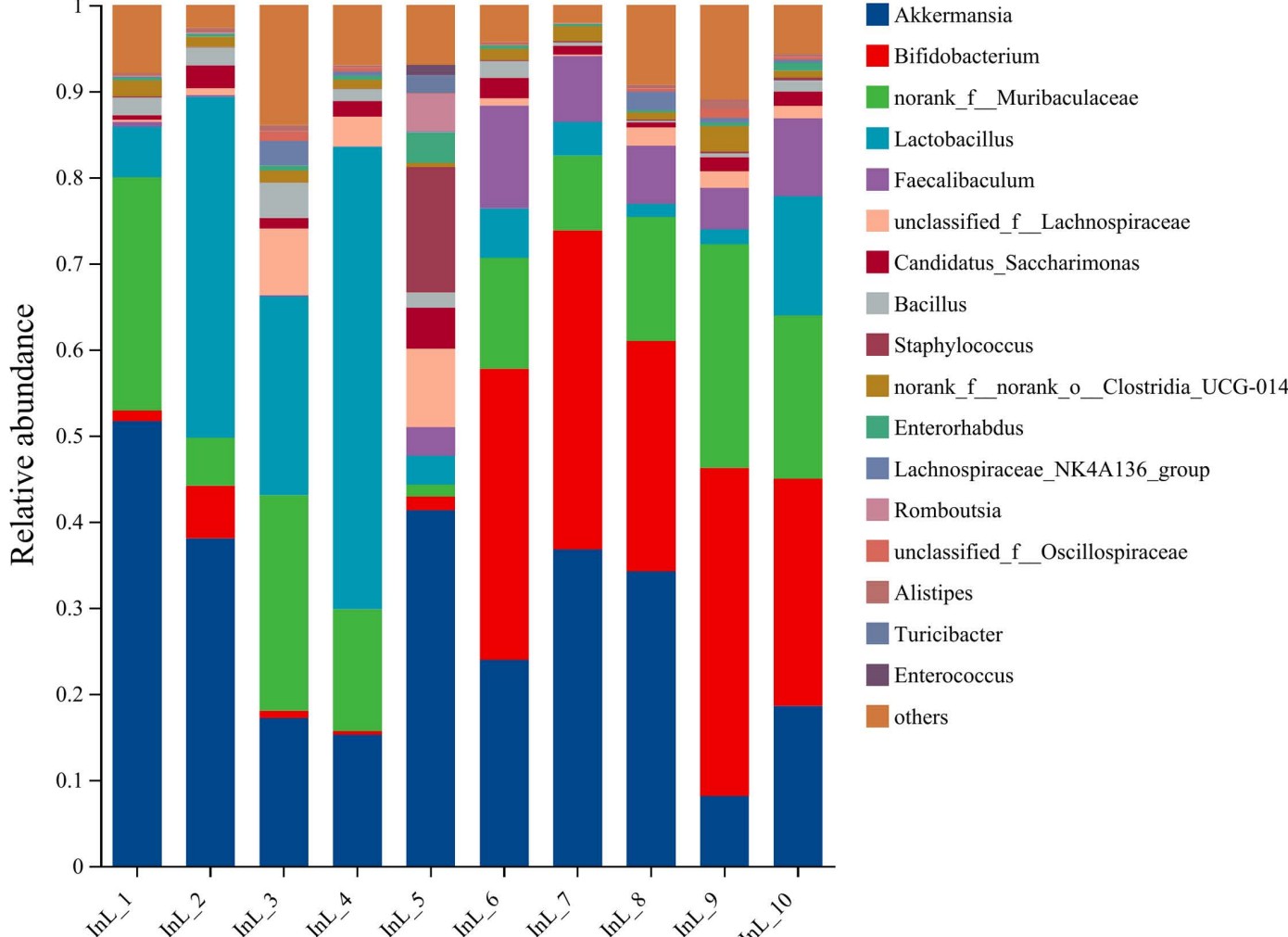

**Fig 18. Genus-level InL group abundance in inulin-treated mice gut microbiota(n =10).**

**Effect of inulin on serum appetite hormone levels in mice.** NPY is an appetite-enhancing neuropeptide that increases appetite and food intake, primarily through its action on the appetite-regulating centre of the hypothalamus. In contrast, PYY is an appetite-suppressing

## Kruskal-Wallis H test bar plot on Genus level

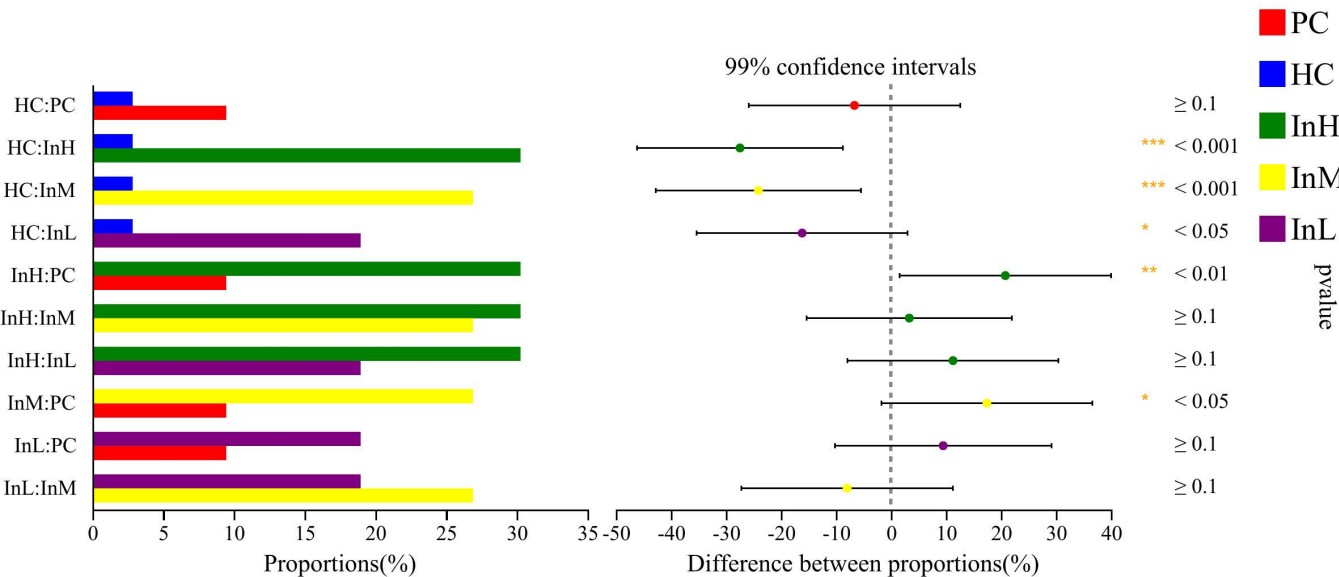

**Fig 19. Inulin-modulated bifidobacterium abundance in functional food evaluation (n=10).** Note: the horizontal coordinates of the left graph indicate the percentage of abundance of a species in each group, while the vertical coordinates indicate the grouping category for the two-by-two comparisons. The right graph indicates the proportion of differences in species abundance within the confidence interval set, with P-values displayed on the right. The symbol *indicate a significant difference of P < 0.05, while the symbol**indicate a highly significant difference of P < 0.01 and ***indicate an extremely significant difference of P < 0.001 compared with the two groups.

## Kruskal-Wallis H test bar plot on Genus level

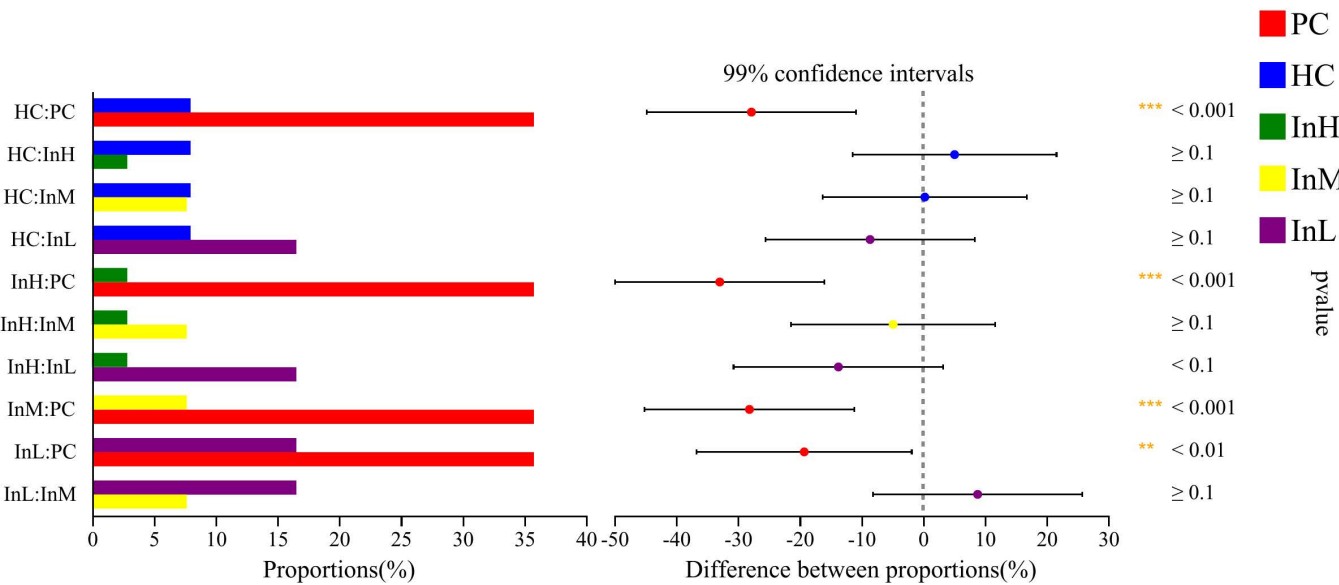

**Fig 20. Inulin-modulated lactobacillus abundance in functional food evaluation (n =10).** Note: Legends follow the same definitions as in Fig 19.

## Kruskal-Wallis H test bar plot on Genus level

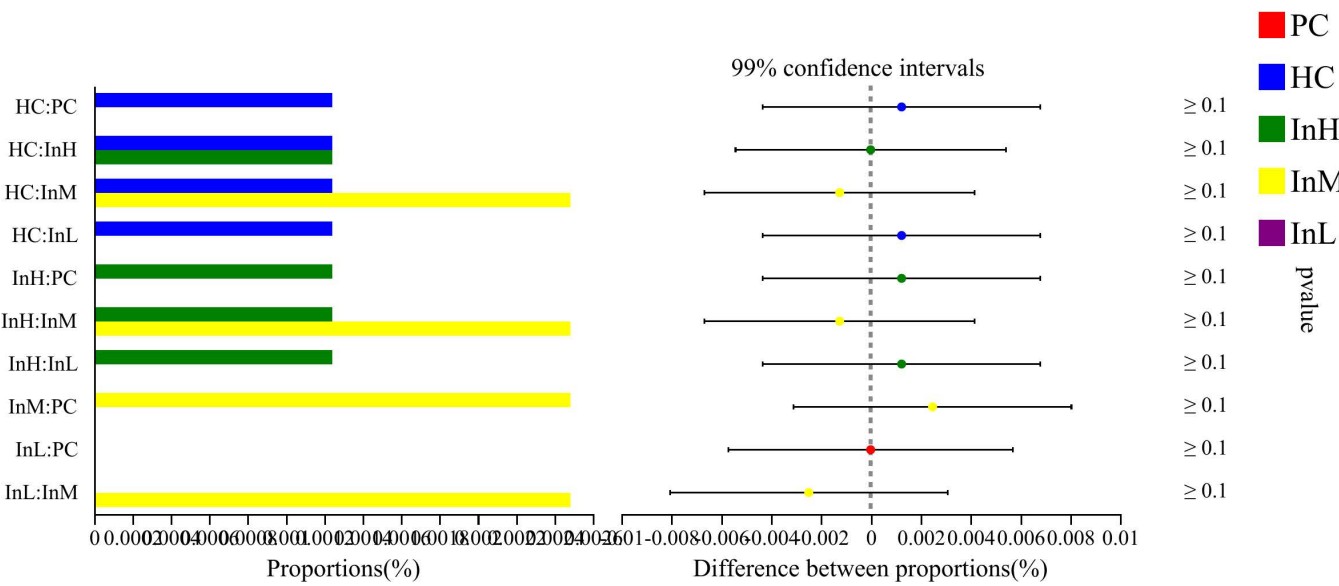

**Fig 21. Inulin-modulated escherichia-shigella abundance in functional food evaluation (n =10).** Note: Legends follow the same definitions as in Fig 19.

## Kruskal-Wallis H test bar plot on Genus level

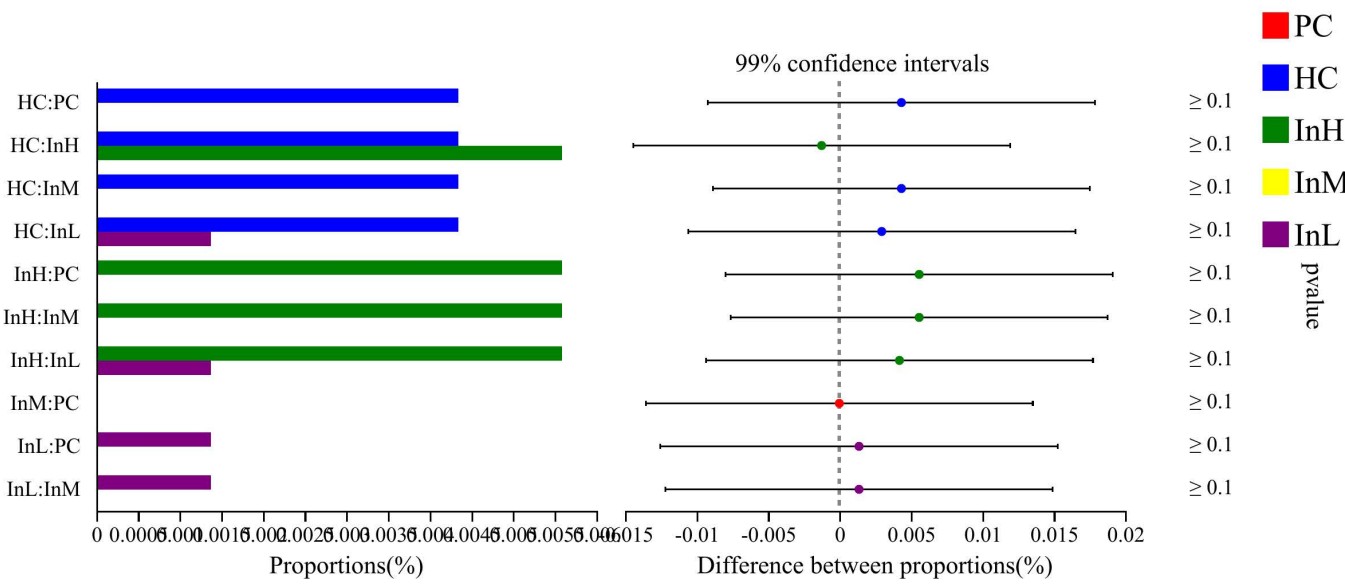

**Fig 22. Inulin-modulated enterococcus abundance in functional food evaluation (n =10).** Note: Legends follow the same definitions as in Fig 19.

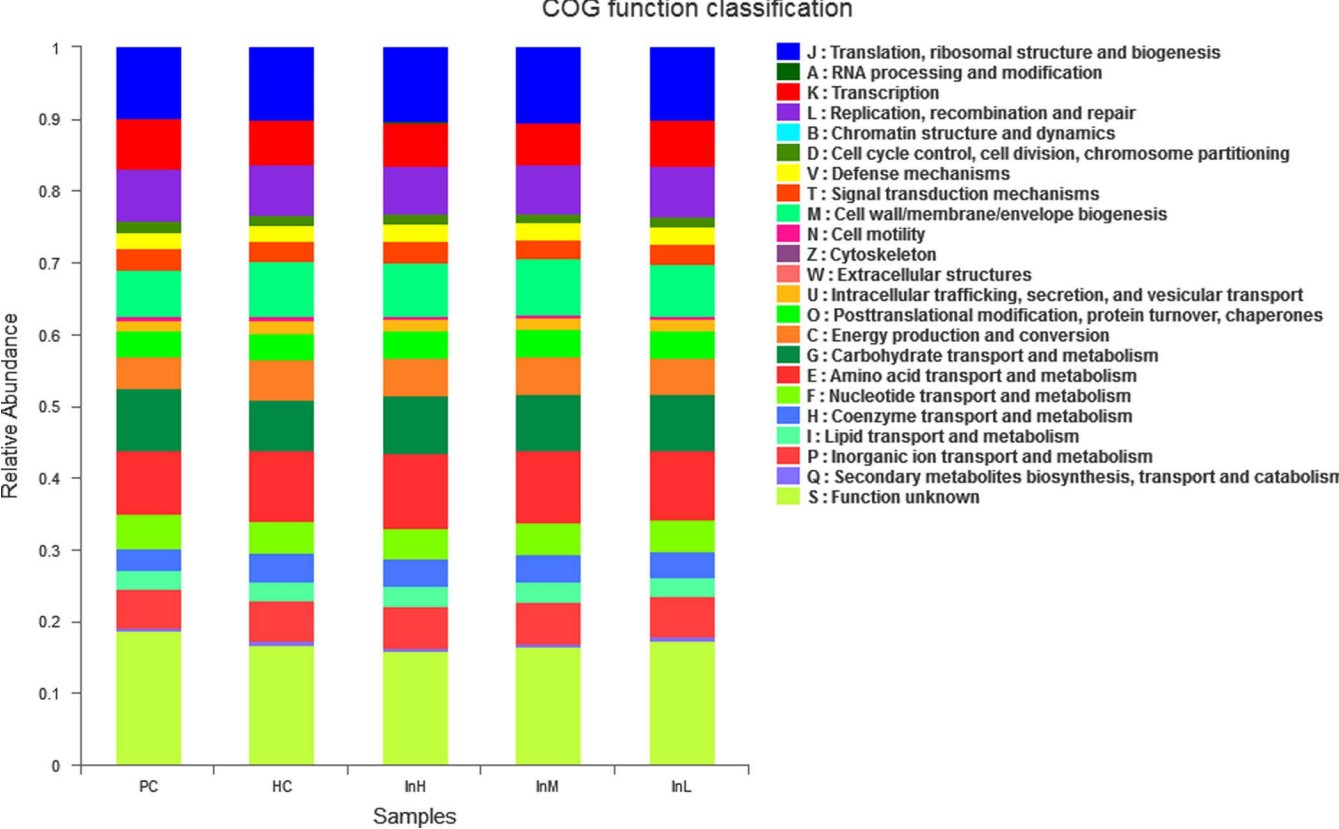

**Fig 23. Functional feature abundance of inulin-treated mice gut microbiota (n =10).**

**Table 7. Function features abundance of gut microbiota of mice under the influence of inulin ($\bar{x}$, n=10).**

| Function | The value of abundance | | | | |
|---|---|---|---|---|---|
| | PC | HC | InH | InM | InL |
| Translation, ribosome structure and biosynthesis | 1,501,697.82 | 851,630.41*** | 1068,808.78***## | 1090,040.32***# | 876,968.93*** |
| Transcription | 1,100,480.93 | 503,715.94*** | 658,079.47***# | 719,520.11** | 491,001.34*** |
| Replication, recombination and repair | 1,113,579.97 | 548,969.94*** | 734,476.08***## | 757,459.32***# | 577,921.15*** |
| Energy production and conversion | 708,362.66 | 439,929.80*** | 576,721.65*## | 550,397.14* | 433,349.27*** |
| Carbohydrate transport and metabolism | 1,301,316.86 | 656,875.73*** | 756,031.66*** | 872,550.63** | 662,779.04*** |
| Amino acid transport and metabolism | 1,385,524.83 | 856,138.10*** | 1036,945.62***## | 1,052,331.38***## | 855,584.93*** |
| Nucleotide transport and metabolism | 733,500.82 | 351,010.08*** | 455,728.30***# | 483,282.38***# | 374,716.59*** |
| Lipid transport and metabolism | 389,564.59 | 221,559.56*** | 284,507.15***## | 286,374.27***# | 232,294.26*** |
| Inorganic ion transport and metabolism | 838,153.57 | 480,366.16*** | 590,913.86***## | 626,465.57***## | 493,713.05*** |
| Secondary metabolites biosynthesis, transport and catabolism | 62,932.47 | 53,063.40 | 52,141.84** | 52,512.39** | 48,631.06** |

Note: 1. The asterisk (*) indicates a statistically significant difference (P < 0.05), while the double asterisk (**) indicates a highly statistically significant difference (P < 0.01) and the triple asterisk (***) indicates an extremely statistically significant difference (P < 0.001) between the PC group and the other groups.2. The symbol # indicates a significant difference of P < 0.05, while the symbol ## indicates a highly significant difference of P < 0.01 compared to the HC group.

neuropeptide that decreases appetite and food intake by inhibiting the appetite-regulating centre in the hypothalamus[31]. Therefore, the reduction in NPY concentration and elevation in PYY concentration in the plateau environment collectively resulted in the suppression of appetite in mice. Compared to the HC group, mice in the inulin group exhibited elevated serum NPY concentrations and diminished PYY concentrations. This indicates that inulin may enhance appetite suppression in a plateau environment by regulating the concentration of serum appetite-regulating hormones. It seems plausible to suggest that the regulation of serum appetite hormones by inulin may be closely related to its regulatory effect on the intestinal flora. The secretion patterns of intestinal hormones, including NPY, PYY, ghrelin, insulin, and IGF-1, can be influenced by intestinal flora. Inulin may alter the secretion pattern of intestinal hormones by promoting the growth and reproduction of beneficial bacteria. Specifically, SCFA produced by inulin fermentation may affect the secretion of NPY and PYY by activating GPR41 and GPR43 receptors in the intestine. Furthermore, inulin may indirectly regulate gut hormone secretion by influencing pH, osmolality, and nutrient metabolism within the gut. In addition to directly influencing gut hormone secretion, inulin may regulate appetite through neurotransmitter action. A close connection exists between neurones in the gut and the central nervous system (CNS), referred to as the "gut-brain axis[32]". Neurotransmitters and hormones in the gut can be transmitted to the brain via the gut-brain axis, thereby affecting the function of appetite regulation centres, such as the hypothalamus. Metabolites such as SCFA, produced by the fermentation of inulin, may be transmitted to the brain via the gut-brain axis, thereby affecting neuronal activity within the hypothalamus. Such neuronal activities may influence appetite and feeding behaviour by modulating the secretion of neuropeptides, including NPY and PYY. Furthermore, inulin may indirectly influence appetite by modulating the activity of other neurotransmitters within the gut, including 5-hydroxytryptamine and gamma-aminobutyric acid. In addition to NPY and PYY, insulin-like growth factor (IGF-1) may also be involved in appetite regulation by inulin. Insulin-like growth factor 1 (IGF-1) is a growth factor secreted by the liver and intestinal cells[33]. It plays various roles, including promoting cell proliferation and differentiation, regulating energy metabolism, and influencing appetite. It is possible that inulin influences the secretion and activity of IGF-1 by improving the structure of intestinal flora. Specifically, the SCFAs produced by inulin fermentation may affect the secretion and activity of IGF-1 by activating GPR43 receptors in the intestine. This effect may further modulate appetite and feeding behaviour in mice.

**The impact of inulin on the intestinal flora of mice.** Sequencing of intestinal flora in this study revealed that the HC group of mice exhibited a reduction in the diversity of intestinal flora, a decline in the number of beneficial bacteria, and an increase in the number of harmful bacteria following their hasty entry into the plateau environment. These changes may be attributed to several factors, including hypoxia, cold, and elevated energy demand resulting from the plateau environment. The hypoxic environment may have inhibited the growth and reproduction of beneficial bacteria in the intestinal tract, whereas the cold and increased energy demand may have promoted the growth and reproduction of harmful bacteria. Such alterations in the structure of the intestinal flora may have influenced the intestinal health and appetite regulation of the mice. The regulatory effects of inulin on intestinal flora were relatively higher in the inulin group than in the HC group, with a corresponding increase in the diversity of intestinal flora and number of beneficial bacteria. This indicates that inulin has the potential to enhance the structure of the intestinal flora in a plateau environment. The regulatory effect of inulin on intestinal flora may be closely related to its properties as a dietary fibre. Inulin can be fermented and used by beneficial bacteria (e.g., Bifidobacterium and Lactobacillus) in the intestinal tract to produce metabolites, such as SCFA. First,

SCFA provide energy and nutrients to intestinal cells, thereby promoting their growth and repair. Second, SCFA can reduce the PH value within the intestinal tract, thereby creating an environment that is less conducive to the growth of harmful bacteria. Third, SCFA can facilitate intestinal peristalsis and the excretion of intestinal contents, thereby reducing the residence time of harmful substances within the intestinal tract. These effects collectively ensure the maintenance of normal physiological functions in the intestinal tract. Furthermore, inulin facilitates intestinal peristalsis, accelerates the discharge of intestinal contents, and reduces the residence time of harmful substances within the intestinal tract[34]. These effects collectively contribute to the maintenance of equilibrium and well-being of the intestinal flora. Finally, there is a close interaction between gut flora and appetite regulation. On the one hand, the composition of the gut flora can influence the secretion pattern of gut hormones, including those that are appetite-related, such as NPY, PYY, ghrelin, and IGF-1. These hormones are transmitted to the brain via the gut-brain axis, where they affect the function of appetite regulation centres, including the hypothalamus and brainstem. Conversely, the functional state of the appetite regulatory centres also influences the structure and function of the gut flora. For example, when mice are in a state of starvation, the hypothalamus releases pro-appetite neuropeptides (e.g., NPY), which may be transmitted to the intestines via the gut-brain axis, promoting the growth and reproduction of beneficial bacteria. Therefore, a complex feedback mechanism exists between gut flora and appetite regulation. It is possible that inulin influences the proper functioning of this feedback mechanism by improving the structure of intestinal flora. In particular, metabolites such as SCFA produced by inulin fermentation may affect neuronal activities within the hypothalamus via the gut-brain axis, which in turn modulates the secretion of neuropeptides such as NPY and PYY. This modulatory effect may have exerted an additional influence on the appetite and feeding behaviour of the mice.

## Conclusion

The hypoxic environment of an acute plateau alters the composition of microbial diversity in the intestinal tract, as well as the abundance values of functional characteristics related to energy metabolism, secondary metabolite synthesis, and transport of the gut microbiota. This, in turn, affects the secretion levels of appetite hormones in various secretory cells in the gastrointestinal tract. Following inulin ingestion, the body can regulate the gut microbiota by increasing the relative abundance of Bifidobacterium and other beneficial bacteria in the intestinal tract of the mice. Furthermore, this process improves the secretion of appetite hormones under conditions of rapid entry to the plateau. Appetite hormones transmit the biological information of appetite regulation to the hypothalamus appetite control centre through the gut-brain axis, thereby alleviating the negative effects on the gut microbiota and food intake of mice after a rapid entry to the plateau. The present study demonstrated that inulin was capable of enhancing appetite suppression in mice through the regulation of intestinal flora structure, improvement of the intestinal environment, and modulation of serum appetite hormone concentration in a multitude of ways within the context of an acute plateau environment. These findings provide a robust scientific foundation for the use of inulin in the context of plateau health. Nevertheless, the present study had some limitations. As the present study was conducted exclusively on mice, future research should investigate the effects of inulin in humans, particularly the modulation of human appetite and intestinal health in a plateau environment. This will provide a crucial foundation for the advancement of inulin as a health supplement.

In conclusion, the findings of this study provide substantial evidence to support the application of inulin in plateau health. Further studies should investigate the mechanism of action

of inulin in greater depth, improve intervention strategies, and conduct human trials to verify its practical effects. This will facilitate the development of innovative approaches to nutritional interventions and health promotion in plateau environments.

## Supporting information

**S1 File. Microbial diversity-PCR formal Lab report (Gel Photos).** Raw gels from agarose gel electrophoresis of PCR products from feces of inulin-treated mice.
(PDF)

**S1 Data. Raw gut microbiota sequencing Data.** The dataset includes multi-comparison analysis results and abundance measurements across experimental groups.
(RAR)

## Acknowledgment

We would like to express our sincere gratitude to Dr. Yuqi Gao and Editor Nie Zhou for their invaluable guidance and support throughout this research. At the same time, thank our families for their unwavering encouragement and patience during the course of this study.

## Author contributions

**Conceptualization:** Xiaoli Li, Shengcai Wu, Xiaonan Chen, Jian Chen, Bao Liu.

**Data curation:** Xiaoli Li, Xiaonan Chen, Bao Liu, XianShuai Liang.

**Formal analysis:** Xiaoli Li, Shengcai Wu.

**Investigation:** Xiaoli Li.

**Methodology:** Xiaoli Li, Shengcai Wu, Xiaonan Chen, Jian Chen, Bao Liu, XianShuai Liang.

**Supervision:** Xiaoli Li, Shengcai Wu, Jian Chen, Bao Liu, XianShuai Liang.

**Writing – original draft:** Xiaoli Li, Shengcai Wu, Xiaonan Chen.

**Writing – review & editing:** Xiaoli Li, Shengcai Wu, Xiaonan Chen.

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
