## [Decision Letter · Decision Letter 0]

3 Sep 2024

PONE-D-24-29210Inulin promotes appetite in mice by regulating the gut microbiota under conditions of rapid entry to the plateauPLOS ONE

Dear Dr. Li,

Thank you for submitting your manuscript to PLOS ONE. After careful consideration, we feel that it has merit but does not fully meet PLOS ONE’s publication criteria as it currently stands. Therefore, we invite you to submit a revised version of the manuscript that addresses the points raised during the review process.

Additional Editor Comments:

Thank you for you submission to PLOS ONE. While your manuscript has merit, the reviewer and I noted several issues that require attention to prior to consideration for approval. My additional comments are listed here with the reviewer's at the end of this letter. Overall the reviewer and I found the research interesting and valuable for PLOS ONE readers but major revisions required.

1. Additional careful proofread is required as there are spelling and grammar errors that need corrected.

2. I do not see that false discovery rate correction was performed for the microbiome evaluations - please perform statistical analysis with FDR correction.

3. Please describe in the methods how the mice were housed and in how many cages they were housed in. Caging effect may be a large confounder of the data.

4. Include dot plot graphs so that the reader and reviewer may see the individual distribution of cohort animals.

We look forward to receiving your revised manuscript.

Kind regards,

Ryan M. Thomas, MD

Academic Editor

PLOS ONE

4. Please amend your manuscript to include your abstract after the title page.

6. Please include a separate caption for each figure in your manuscript.

7. PLOS ONE now requires that authors provide the original uncropped and unadjusted images underlying all blot or gel results reported in a submission’s figures or Supporting Information files. This policy and the journal’s other requirements for blot/gel reporting and figure preparation are described in detail at https://journals.plos.org/plosone/s/figures#loc-blot-and-gel-reporting-requirements and https://journals.plos.org/plosone/s/figures#loc-preparing-figures-from-image-files. When you submit your revised manuscript, please ensure that your figures adhere fully to these guidelines and provide the original underlying images for all blot or gel data reported in your submission. See the following link for instructions on providing the original image data: https://journals.plos.org/plosone/s/figures#loc-original-images-for-blots-and-gels.  

Reviewers' comments:

Reviewer's Responses to Questions

**Comments to the Author**

1. Is the manuscript technically sound, and do the data support the conclusions?

Reviewer #1: Yes

2. Has the statistical analysis been performed appropriately and rigorously? 

Reviewer #1: I Don't Know

3. Have the authors made all data underlying the findings in their manuscript fully available?

Reviewer #1: Yes

4. Is the manuscript presented in an intelligible fashion and written in standard English?

Reviewer #1: Yes

5. Review Comments to the Author

Reviewer #1: Editor Comments:

The manuscript presents an interesting study on the effects of inulin on appetite modulation in mice, focusing on gut microbiota changes under rapid entry to plateau conditions. The concept is well-founded, and the methodology appears robust. However, several areas require revision and clarification to enhance the manuscript's clarity and scientific rigor.

Specific Comments:

1. Introduction: The introduction should better contextualize the research within existing literature. Please expand on previous studies that have investigated the effects of inulin on gut microbiota and appetite, particularly in non-plateau conditions, to strengthen the rationale for this study.

2. Data presentation: Some figures and tables are very complex. Simplify them if possible.

3. Discussion: The discussion section needs further improvement. Explain how the results are consistent with the initial hypothesis. Discuss the implications of these findings for using inulin as a functional food, especially under the unique conditions of rapid plateau entry. The discussion should also include a section on study limitations. Address potential biases and generalizability of the results to other populations or species.

4. References: Verification and Formatting: Please review references 1 to 11 for relevance and accuracy, ensuring they are updated and formatted according to the journal’s guidelines.

6. PLOS authors have the option to publish the peer review history of their article (what does this mean? ). If published, this will include your full peer review and any attached files.

**Do you want your identity to be public for this peer review?** For information about this choice, including consent withdrawal, please see our Privacy Policy .

Reviewer #1: **Yes: ** Dr. Mai Albaik

---

## [Author Response · Author response to Decision Letter 0]

13 Nov 2024

Dear Ryan M. Thomas,

I have examined the comments you provided and have incorporated the following four main elements into the manuscript:

1. The figures have been removed from the manuscript.

2. All authors and their respective affiliations are listed in the main document, and the identity of the corresponding author is clearly indicated.

3. The separately loaded table files have been removed.

I would be grateful if you could review the revised manuscript and provide me with your invaluable feedback on this revised version.

Kind regards,

Li Xiaoli

---

## [Decision Letter · Decision Letter 1]

11 Mar 2025

PONE-D-24-29210R1Inulin promotes appetite in mice by regulating the gut microbiota under conditions of rapid entry to the plateauPLOS ONE

Dear Dr. Li,

Thank you for submitting your manuscript to PLOS ONE. After careful consideration, we feel that it has merit but does not fully meet PLOS ONE’s publication criteria as it currently stands. Therefore, we invite you to submit a revised version of the manuscript that addresses the points raised during the review process.

The authors have clearly addressed the comments from the first round of reviews. A second round of review suggests a minor revision is required. I also advise the authors to update the background with rescent publications that came after the original submission. 

We look forward to receiving your revised manuscript.

Kind regards,

Yash Gupta, Ph.D.

Academic Editor

PLOS ONE

Journal Requirements:

Additional Editor Comments:

Reviewers have raised deficiencies that are to be corrected.

Reviewers' comments:

Reviewer's Responses to Questions

**Comments to the Author**

1. If the authors have adequately addressed your comments raised in a previous round of review and you feel that this manuscript is now acceptable for publication, you may indicate that here to bypass the “Comments to the Author” section, enter your conflict of interest statement in the “Confidential to Editor” section, and submit your "Accept" recommendation.

Reviewer #2: All comments have been addressed

Reviewer #3: (No Response)

2. Is the manuscript technically sound, and do the data support the conclusions?

Reviewer #2: Yes

Reviewer #3: Yes

3. Has the statistical analysis been performed appropriately and rigorously? 

Reviewer #2: Yes

Reviewer #3: N/A

4. Have the authors made all data underlying the findings in their manuscript fully available?

Reviewer #2: Yes

Reviewer #3: Yes

5. Is the manuscript presented in an intelligible fashion and written in standard English?

Reviewer #2: Yes

Reviewer #3: Yes

6. Review Comments to the Author

Reviewer #2: (No Response)

Reviewer #3: Present work, titled as “Inulin promotes appetite in mice by regulating the gut microbiota under conditions of rapid entry to the plateau”, by Xiaoli Li and co-workers aims to explore role of inulin, a dietary fiber, in regulating gut microbiota with a model study using fifty seven-week-old SPF-grade C57BL/6J male mice. Authors have chosen two different environmental conditions, i.e. regular and rush to plateau to study structural as well as functional characteristics of inulin uptake affecting the gut microbiota. Results demonstrate that inulin has a direct impact on metabolism and secretion of appetite hormones which result in enhanced appetite and body weight by increasing useful gut bacteria.

Authors have systematically introduced the concepts involved in developing the study of inulin in the context of plateau health supplements along with detailed methodology adopted in conducting the study. The manuscript is well-written, highlighting the background of current study, limitations/challenges followed by scope of future actions towards promoting health in plateau environments. The work can be accepted for publication after addressing the following concerns:

Correction:

• Authors mentioned in the abstract that ‘fifty seven-week-old SPF-grade C57BL/6J male mice’ were used for the study but materials and methods section says fifty eight week. Please check and correct this.

• What do NPY and PYY correspond to in the ‘Measurement of appetite hormone content’ section? Explain the significance of these measuring these two contents for better understanding of the readers.

• Which ‘spectrophotometer’ was used to measure the purity and concentration of extracted DNA? Provide the details.

• Provide the reference for ‘four parameter logistic function’.

• Several sentences need to be rephrased for clarity like;

‘Overcoming the adverse effects………………………………………………….…solved urgently.’

‘Furthermore, the mechnanism……………………………………………………inulin was analyzed.’

‘Were/was’ used at several places in the results and discussion section.

‘pH’ written as ‘PH’.

‘Inulin’ should be written in lower case whenever it appears in the middle of the sentence.

7. PLOS authors have the option to publish the peer review history of their article (what does this mean? ). If published, this will include your full peer review and any attached files.

**Do you want your identity to be public for this peer review?** For information about this choice, including consent withdrawal, please see our Privacy Policy .

Reviewer #2: No

Reviewer #3: No

---

## [Author Response · Author response to Decision Letter 1]

12 Mar 2025

Dear Editor and Reviewers,

Thank you for your constructive feedback and careful evaluation of our manuscript titled “Inulin promotes appetite in mice by regulating the gut microbiota under conditions of rapid entry to the plateau.” We have thoroughly addressed all the concerns raised and incorporated the suggested revisions. Below are our point-by-point responses:

1.Correction: Inconsistency in mouse age

Reviewer’s comment: The abstract states “fifty seven-week-old mice,” while the Materials and Methods section mentions “fifty eight-week-old mice.”

Response: We sincerely apologize for this oversight. The correct age of the mice is seven-week-old, as stated in the abstract. The typographical error in the Materials and Methods section has been corrected (Page 3, Line 78).

2.Clarification of NPY and PYY

Reviewer’s comment: Explain the abbreviations NPY and PYY and their significance in appetite regulation.

Response: We have revised the “Measurement of appetite hormone content” section to clarify:

Neuropeptide Y(NPY, an appetite-stimulating hormone). Peptide YY(PYY, an appetite-suppressing hormone).This explanation has been added to the manuscript (Page 6, Lines 68–69).

Measuring these hormones helps elucidate how inulin modulates appetite via gut microbiota-mediated pathways. This explanation has been added to the manuscript (Page 6, Lines 112–114).

3.Spectrophotometer details

Reviewer’s comment: Specify the spectrophotometer used for DNA analysis.

Response: The purity and concentration of the extracted DNA were measured using an AOELAB V-1200 spectrophotometer made by Xiangyi Instruments Co. Ltd (Shanghai, China).

This detail has been included in the Methods section (Page 4, Line 73-76).

4.Reference for the four-parameter logistic function

Reviewer’s comment: Provide a reference for the four-parameter logistic function.

Response: We have added the requested reference (DeLean et al., 1978) as citation [27] in the revised manuscript (Page 9, Line 191).

5. Language and formatting revisions

Reviewer’s comment: Rephrase ambiguous sentences, correct grammatical errors, and standardize formatting (e.g., “pH,” lowercase “inulin”).

Response: All suggested revisions have been implemented:

Ambiguous sentences (e.g., “Overcoming the adverse effects…solved urgently”) were rephrased for clarity.

Grammatical inconsistencies (e.g., “were/was”) and formatting errors (e.g., “PH” → “pH”) were corrected.

“Inulin” is now consistently lowercase in mid-sentence instances.

We appreciate the reviewer’s meticulous attention to these details.

Additional Changes

All figures, tables, and supplementary materials were reviewed for consistency, and minor typographical errors were corrected throughout the manuscript.

We believe these revisions have significantly strengthened the manuscript. Thank you again for your valuable feedback. Please let us know if further clarifications or adjustments are required.

Sincerely,

Xiaoli Li (on behalf of all authors)

---

## [Editor Report · Decision Letter 2]

16 Mar 2025

Inulin promotes appetite in mice by regulating the gut microbiota under conditions of rapid entry to the plateau

PONE-D-24-29210R2

Dear Dr. Li,

We’re pleased to inform you that your manuscript has been judged scientifically suitable for publication and will be formally accepted for publication once it meets all outstanding technical requirements.

Kind regards,

Yash Gupta, Ph.D.

Academic Editor

PLOS ONE
---

## [Editor Report · Acceptance letter]

PONE-D-24-29210R2

PLOS ONE

Dear Dr. Li,

I'm pleased to inform you that your manuscript has been deemed suitable for publication in PLOS ONE. Congratulations! Your manuscript is now being handed over to our production team.

Kind regards,

on behalf of

Dr. Yash Gupta

Academic Editor

PLOS ONE